# LaDIVA: A neurocomputational model providing laryngeal motor control for speech acquisition and production

**Hasini R. Weerathunge**[1,2]\*, **Gabriel A. Alzamendi**[3,4], **Gabriel J. Cler**[5], **Frank H. Guenther**[1,2], **Cara E. Stepp**[1,2,6], **Matías Zañartu**[3]

**1** Department of Biomedical Engineering, Boston University, Boston, Massachusetts, United States of America, **2** Department of Speech, Language, and Hearing Sciences, Boston University, Boston, Massachusetts, United States of America, **3** Department of Electronic Engineering, Universidad Técnica Federico Santa María, Valparaíso, Chile, **4** Institute for Research and Development on Bioengineering and Bioinformatics (IBB), CONICET-UNER, Oro Verde, Argentina, **5** Department of Speech & Hearing Sciences, University of Washington, Seattle, Washington, United States of America, **6** Department of Otolaryngology-Head and Neck Surgery, Boston University School of Medicine, Boston, Massachusetts, United States of America

\* hasiniw@bu.edu

**Data Availability Statement:** The source code of LaDIVA model is available at https://sites.bu.edu/guentherlab/software/diva-source-code/.

## Abstract

Many voice disorders are the result of intricate neural and/or biomechanical impairments that are poorly understood. The limited knowledge of their etiological and pathophysiological mechanisms hampers effective clinical management. Behavioral studies have been used concurrently with computational models to better understand typical and pathological laryngeal motor control. Thus far, however, a unified computational framework that quantitatively integrates physiologically relevant models of phonation with the neural control of speech has not been developed. Here, we introduce *LaDIVA*, a novel neurocomputational model with physiologically based laryngeal motor control. We combined the DIVA model (an established neural network model of speech motor control) with the extended body-cover model (a physics-based vocal fold model). The resulting integrated model, LaDIVA, was validated by comparing its model simulations with behavioral responses to perturbations of auditory vocal fundamental frequency ($f_o$) feedback in adults with typical speech. LaDIVA demonstrated capability to simulate different modes of laryngeal motor control, ranging from short-term (i.e., *reflexive*) and long-term (i.e., *adaptive*) auditory feedback paradigms, to generating prosodic contours in speech. Simulations showed that LaDIVA's laryngeal motor control displays properties of motor equivalence, i.e., LaDIVA could robustly generate compensatory responses to reflexive vocal $f_o$ perturbations with varying initial laryngeal muscle activation levels leading to the same output. The model can also generate prosodic contours for studying laryngeal motor control in running speech. LaDIVA can expand the understanding of the physiology of human phonation to enable, for the first time, the investigation of causal effects of neural motor control in the fine structure of the vocal signal.

**Funding:** This research was supported by the National Institutes of Health (NIH; https://www.nih.gov/) National Institute on Deafness and Other Communication Disorders (NIDCD;https://www.nidcd.nih.gov/) through Grants No. P50 DC015446 (C.E.S, M.Z), F31 DC014872(G.C), F32 DC017637 (G.C), R01 DC016270(C.E.S, F.H.G), and R01 DC002852(F.H.G), and by the Agencia Nacional de Investigación y Desarrollo (ANID;https://www.anid.cl/) through Grants Nos. FONDECYT 1191369 (M. Z) and BASAL FB0008(M.Z). The funders had no role in study design, data collection and analysis, decision to publish, or preparation of the manuscript. The content is solely the responsibility of the authors and does not necessarily represent the official views of the NIH.

**Competing interests:** The authors have declared that no competing interests exist.

## Author summary

With the incorporation of a physiologically relevant vocal fold model into a computational speech motor control framework, LaDIVA is a neurocomputational model that includes realistic laryngeal activity. The proposed model has demonstrated capability to simulate different modes of laryngeal motor control, ranging from short-term (i.e., *reflexive*) and long-term (i.e., *adaptive*) auditory feedback paradigms, to generating prosodic contours in speech. LaDIVA displays properties of motor equivalence, i.e., it robustly generates similar compensatory responses to auditory perturbations for all simulations regardless of the variety in initial laryngeal muscle activation levels tested. LaDIVA can be used to expand the understanding of the physiology of human phonation to enable, for the first time, the investigation of causal effects of neural motor control in the fine structure of the vocal signal.

## Introduction

Voice disorders are a common and recurring occupational health disorder, affecting 7% of the U.S. population and causing severe mental, physical, and emotional repercussions, as well as adverse consequences on job performance [1]. Fifty percent of teachers report voice problems during their career [2] and 2.5 billion dollars each year are spent on treatments and sick leave for voice disorders in the U.S. [3]. Many voice disorders are the result of intricate neural and biomechanical impairments that are poorly understood [4]. Effective clinical management of many of these voice pathologies is hampered by the limited knowledge of their etiology and pathophysiological mechanisms, as well as overlapping symptoms among multiple disorders [5,6]. Identification of the pathophysiology of voice impairments and decoupling the neural and biomechanical aspects of the pathology may provide new directions for clinical intervention.

   Speech production is a complex motor skill involving a multitude of neuromuscular executions in respiratory, laryngeal, and articulatory muscles [7]. Simultaneously, the neural motor controllers integrate sensory information from multiple feedback modalities (i.e., auditory and somatosensory feedback) to monitor the accuracy of the acoustic output [8]. Neurocomputational models of speech are designed to systematically unify available knowledge on speech motor control, to simulate behavior, and test hypotheses related to speech motor control architecture [9–12]. Likewise, computational models of voice production have improved our understanding of typical and pathological phonation by providing access to relevant features that are difficult, if not impossible, to directly measure [13–16]. However, current models do not yet integrate physiologically relevant models of phonation into computational models of neural motor control, which would allow for investigation of auditory and somatosensory processing in voice production. Thus, despite the contribution of these models to enhance the clinical assessment of vocal function, their capacity to study laryngeal motor control is lacking.

   Neurocomputational models use mathematical representations and/or neural networks to abstract components of brain function in a straightforward manner [17]. Of current biologically plausible neural network models of speech production [18,19], the *Directions into Velocities of Articulators* model (DIVA; [8, 20]) is possibly the most thoroughly defined and physiologically validated neurocomputational model for articulatory speech motor control to date. The DIVA model incorporates detailed neuroanatomical and neurophysiological information into a unified platform, making it well-suited to generate simulations of speech production for meaningful, quantitative comparisons with experimental observations [8,20–22].

DIVA is a neural network model whose components correspond to regions of the cerebral cortex and cerebellum, including premotor, motor, auditory, and somatosensory cortical areas. Although DIVA does not provide a complete account of the cortical and cerebellar mechanisms involved, the mathematical model has been able to replicate behavioral and neural responses via simulations of simple syllable productions. The model uses decades of behavioral and neuroimaging data to tie the components of the mathematical framework to neural substrates particularly related to articulation. The DIVA model was developed to replicate human speech produced at behavioral, neurological, and developmental levels [8,11,20,23,24]. It consists of a hybrid control system combining three main control components related to speech articulation: 1) a feedforward controller that utilizes internally stored motor programs to produce sound, 2) an auditory feedback controller that detects errors between actual acoustic output and acoustic feature targets, and 3) a somatosensory feedback controller that detects errors between actual somatosensory (i.e., kinesthetic or proprioceptive) output and somatosensory feature targets. The DIVA model has been successfully used in conjunction with brain imaging and behavioral experiments to refine our understanding of the neural control of speech and generate hypotheses for further behavioral studies [8,12,23–25]. However, the neuromuscular control of laryngeal function is an important aspect of speech production that is not addressed in the DIVA model [25]. Key acoustic features of voice production, such as vocal fundamental frequency ($f_o$), vocal sound pressure level (SPL), and voice signal quality are modulated via laryngeal motor control in humans [26,27], and prior research suggests that disorders such as spasmodic dysphonia, vocal hyperfunction, spastic dysarthria, and vocal tremor may be related to abnormal laryngeal motor control [28–33]. Incorporating laryngeal motor control to models of speech motor control will enhance the ability to understand these disorders.

Characterizing laryngeal function in phonation requires a detailed understanding of the underlying physiology, biomechanics, and neuromuscular control. The neural motor control and biomechanical mechanisms of typical and disordered laryngeal function involved in vocal production, and the interaction between them are poorly understood, partly due to the invasive nature of assessment and characterization of laryngeal function (i.e., laryngeal electromyography: [34–39], laryngeal endoscopy: [40,41], contact pressure probes:[42]). The geometry and mechanical properties of the vocal folds (VFs) exhibit a great deal of variability as a function of intrinsic muscle activation, voicing conditions, sex, age, and individual anatomical features, and complex physical interactions occur between VF tissue, airflow, and sound [43,44]. Thus, it is crucial to investigate these components and their variability to better understand laryngeal function in typical and pathologic conditions. Low-order models of phonation such as lumped-element VF models with acoustic and aerodynamic modules offer a simple yet physiologically relevant framework for investigating biomechanical aspects of voice production while remaining computationally efficient [45]. For instance, the well-established body-cover model (BCM; [46–48]) can control supra and subglottal tract features such as subglottal pressure (Ps), and neuromuscular activation of the cricothyroid (CT), thyroarytenoid (TA), lateral cricothyroid (LCA), and posterior cricothyroid (PCA; [48]) intrinsic laryngeal muscles [47–49]. Models extending the long-standing BCM have been broadly used to predict complex physical phenomena in voice production in humans, such as irregular vibration in VF nodules [50], contact mechanics [13,51], compensatory responses in vocal hyperfunction [14,49], chaotic vibration in unilateral VF paralysis [52,53], and dynamic mechanisms of atypical vocal fold vibration in vocal tremor [54]. As higher order three-dimensional finite element models [43,54–56] are computationally demanding to account for neural motor control effects, low-order models are most suitable to comprehensive parametric stimulations of laryngeal motor control. The extended body cover model selected in the current study (hereafter referred to as extended BCM; [49]) is an extension of the BCM by Story and Titze (46), with the addition of

a posterior glottal opening, and model parameters selected to simulate a male modal voice using muscle activation principles defined in Titze and Story (48). See the original paper for the equations for the motion of vocal folds in the extended BCM model [46]. The extended BCM was specially selected considering its simplicity as well as scalability for future iterations of LaDIVA. In summary, combining a low-order biomechanical model with models of neural laryngeal motor control will provide an integrated framework to simulate vocal productions that are comparable to outputs of behavioral study data collected from individuals with and without voice disorders [57–60].

Behavioral studies are often conducted with the aim of understanding and quantifying responses of the speech motor control system to erroneous feedback stimuli. Prior research has behaviorally characterized laryngeal motor control with externally imposed short-term (i.e. *reflexive)* and long-term (i.e. *adaptive*) alterations of auditory feedback of laryngeal features (i.e., vocal $f_o$, vocal SPL, duration). Reflexive paradigms targeting vocal $f_o$ (the physical quantity underlying the perceptual feature of pitch) present sudden, unpredictable alterations to auditory $f_o$ feedback to measure the feedback-based error detection, correction, and incorporation to real-time motor production (commonly termed pitch reflex; [61–64]). Adaptive vocal $f_o$ paradigms present predictable, sustained perturbations to auditory $f_o$ feedback over time to measure the speaker's ability to correct errors in subsequent feedforward motor commands [65–67]. Behavioral studies of these types have been performed with those with typical voices [68] as well as speakers with neurological and/or voice disorders (e.g., Parkinson's disease [PD; 69] and vocal hyperfunction [57,70,71]). Studies using auditory feedback perturbations in vocal $f_o$ have indicated that individuals with PD have a higher reliance on the auditory feedback subsystem for vocal motor control [72–75], and impaired auditory-motor integration in terms of feedforward motor control [69]. Similar studies have also provided evidence that individuals with vocal hyperfunction demonstrate potential signs of a motor speech disorder [57,70,71] with difficulty using auditory feedback to update their feedforward control subsystems. Moreover, these studies also observed larger individual variability in the vocal responses of individuals with vocal hyperfunction compared to speakers with typical speech. Overall, the outcomes of these studies suggest there are possible vocal motor control impairments related to vocal $f_o$ in specific populations with voice disorders. These experimental data have been replicated via a simple three-parameter model of speech and vocal production: simpleDIVA [76]. Although simpleDIVA successfully abstracts the DIVA architecture to model both articulatory and laryngeal motor control, the simplification of DIVA to three mathematical equations has its own limitations; namely, it implicitly assumes a laryngeal system that will perfectly achieve commanded movements. A neurocomputational model of laryngeal motor control equipped with biomechanical mechanisms of laryngeal function would allow comprehensive investigation via repeated simulations of numerous experimental conditions, that may be exhaustive to participants during behavioral experimentation, or inaccessible to conduct/assess due to technical limitations in experimentation (e.g., voice quality perturbation, combined perturbations, running speech perturbation). Model simulations of laryngeal motor control would be particularly valuable to address these limitations and to better understand typical and atypical laryngeal motor control behaviors.

In this paper, we introduce the laryngeal DIVA model (*LaDIVA)*, a neurocomputational model that integrates biomechanics of phonation and neural laryngeal motor control in a single computational framework. The novel model is based on two widely accepted existing models, the DIVA model for the neural laryngeal motor control component [8] and the extended BCM [49] for the biomechanical component. The combined model, LaDIVA, is able to dynamically control the intrinsic laryngeal muscles of the extended BCM by assessing the resultant acoustic output based on auditory feedback and feedforward mechanisms. LaDIVA

will significantly expand the capabilities and potential of current VF modeling tools, particularly for investigating voice disorders in which neuropathy related to auditory and/or somatosensory systems is suspected. In addition, LaDIVA can be used to predict variations in underlying feedforward and feedback control mechanisms between typical vocal production and disordered voice production, which could motivate future behavioral study designs to investigate laryngeal function in voice disorders.

In the following sections, we describe the construction of LaDIVA and the initial simulations performed to validate the model using behavioral data. The *Model Overview* summarizes the architecture and the computational implementation of LaDIVA, which currently enables vocal $f_o$ and vocal SPL (the physical quantity underlying the perceptual feature of loudness) control. In the *Results*, we evaluate LaDIVA to show that it can replicate human laryngeal motor control qualitatively and quantitatively. Data collected from a previous behavioral experiment [68] from adults with typical speech were used to validate LaDIVA.

## Model overview

LaDIVA is a neurocomputational laryngeal control model, combining two well-established and physiologically validated models describing different aspects of speech production. DIVA [8], a neurocomputational model of speech, and the extended BCM [49], a biomechanical model of phonation. *Fig 1* illustrates a simplified version of LaDIVA architecture with adapted modules from DIVA shaded in light grey and adapted modules from the extended BCM shaded in dark grey. The controlled variables of the model are listed in *Table 1*.

In DIVA [8], there are multi-dimensional goal regions defined for controlled variables. These regions are termed targets and are expressed in task space and mobility space variables. The relevant task space variables in DIVA required to define laryngeal movement are vocal $f_o$ for auditory tasks pace and vocal SPL for somatosensory task space (see *Table 1*: in LaDIVA, vocal SPL is modified to be an auditory task space variable). The motor planning layer of DIVA contains the speech sound map, an expansive list of speech sounds composed of phonemes, syllables, or word sounds. Each speech sound is mapped to a three-component representation of controlled variables. The first component is an articulatory trajectory (i.e., generally referred to as motor trajectories) that defines the mobility space targets of a specific speech sound (i.e., the information related to the position of a particular articulator; represented in orange in *Fig 1*). The other two are distinct target trajectories in auditory and somatosensory task spaces (represented in purple and green respectively in *Fig 1*). These three types of target trajectories constitute the reference signals for the feedforward controller, auditory feedback controller, and somatosensory feedback controller in DIVA, respectively.

At the beginning of a speech sound production, the **speech sound map** for that sound is activated, leading to a readout from memory of a learnt set of motor target trajectories for that specific sound. The **feedforward controller** compares these motor targets with the *current motor control signal* ($C^i_{motor}(x, t)$; dark orange arrows) and generates a *feedforward motor control signal* ($FF_{motor}(x,t)$; light orange arrows). In the **feedback controllers**, auditory and somatosensory feedback error between the sensory outputs and desired sensory targets are **inversely transformed** (i.e., $J^{-1}(x)$) from tasks space variables to mobility space variables, and the resultant *feedback error correction signals* (i.e., $FB_{aud}(x,t)$ and $FB_{som}(x,t)$; dotted light orange arrows), are incorporated to the feedforward control signal to generate the *subsequent motor control signal* ($C^{i+1}_{motor}(x, t)$; i.e., dark orange arrows). DIVA then uses a **forward transformation** (i.e., $F\{\sim\}$) to generate the subsequent auditory and somatosensory outputs relevant to the motor control signal.

DIVA primarily controls movements of the supralaryngeal vocal tract articulators. These articulators define the shape of the vocal tract "tube", which is used in combination with a

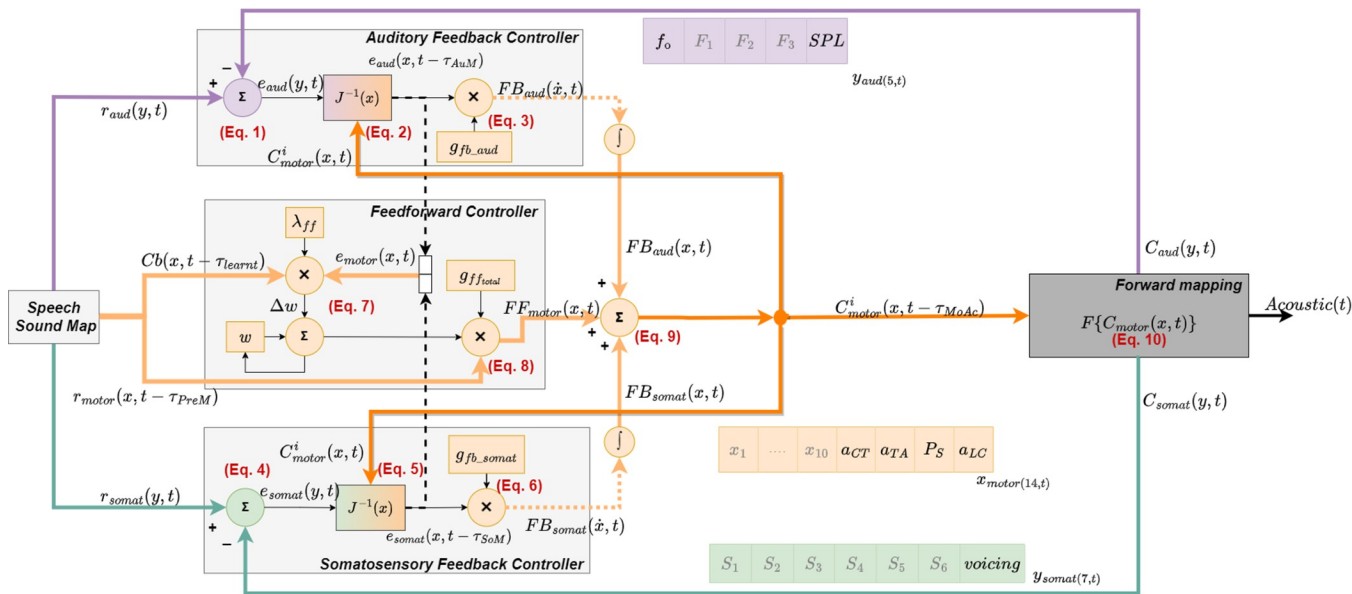

**Fig 1. LaDIVA model architecture.** Modules in light grey are adapted from the DIVA model and the module in dark grey represents the larynx, mathematically modeled via the extended body-cover model for vocal folds. Mobility space variables are denoted by x, and task space variables are denoted by y. Pathways denoted in auditory task space are purple. Pathways denoted in somatosensory task space are green. Pathways denoted by motor mobility space are orange. See *Materials and Methods* for annotated equations.

simplified glottal wave model [77] to convert vocal tract articulatory mobility space trajectories into auditory and somatosensory task space trajectories [78]. Ten different articulatory position dimensions are used to define articulatory mobility space trajectories. The forward transformation of these articulatory trajectories results in sensory trajectories with three auditory task space dimensions (i.e., $F_1$, $F_2$, $F_3$) and seven somatosensory task space dimensions (i.e., six places of constriction in the vocal tract and a glottal constriction). The model also includes three voicing-related output variables (i.e., for VF tension, subglottal pressure, and voicing), but values for these are specified directly by the user rather than computed by the model's feedforward and feedback control system interactions.

**Table 1. LaDIVA Model Parameters.**

| DIVA component → | | | BCM component |
|---|---|---|---|
| **Mobility Space** ($x_{motor}$) | Articulatory dimensions (10) | Ten vocal tract shapes | N/A |
| | Source dimensions (4) | Cricothyroid Muscle Activation ($a_{CT}$), Thyroarytenoid Muscle Activation ($a_{TA}$), Subglottal Pressure ($P_S$) | Input Parameters |
| | | Glottal constriction dimension | Voiced state handled via BCM ($a_{LC}$) |
| DIVA component ← | | | BCM component |
| **Auditory Task Space** ($y_{aud}$) | Source dimensions (2) | Vocal Fundamental Frequency ($f_o$), Vocal Sound Pressure Level (SPL) | Output Parameters (derived via glottal flow signal; $U_g$) |
| | Articulatory dimensions (3) | First Formant ($F_1$), Second Formant ($F_2$), Third Formant ($F_3$) | N/A |
| **Somatosensory Task Space** ($y_{somat}$) | Articulatory dimensions (6) | Place of constriction in vocal tract: Labial, Alveolar/Dental, Palatal, Velar, Uvular, Pharyngeal | N/A |
| | Source dimensions (1) | Glottal constriction dimension | Voiced state handled via BCM ($a_{LC}$) |

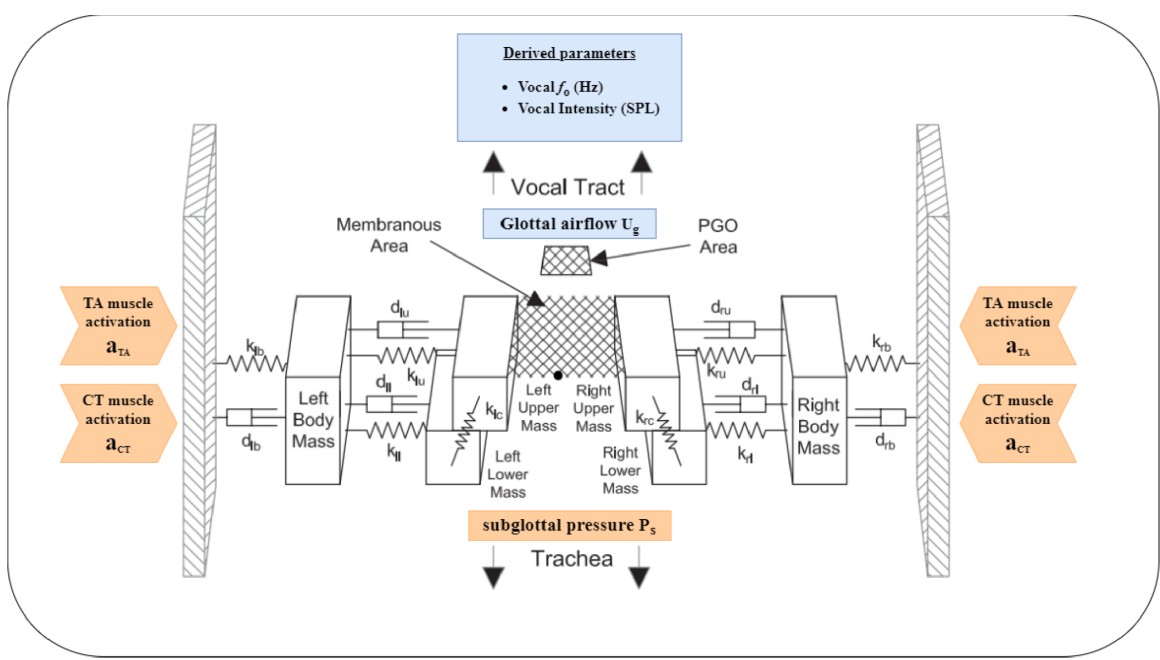

**Fig 2. Extended Body-Cover Model.** Vocal fold mobility variables input from or output to the DIVA model are shown in orange and blue shaded regions, respectively. Reproduced from Zañartu, Galindo [49] (https://doi.org/10.1121/1.4901714), with the permission of the Acoustical Society of America.

In contrast, in LaDIVA, the extended BCM is used to convert VF mobility space trajectories (i.e., laryngeal trajectories) to changes in auditory and somatosensory task space trajectories. Three different VF mobility space dimensions are used, namely, 1) activation level of cricothyroid muscle ($a_{CT}$), 2) activation level of thyroarytenoid muscle ($a_{TA}$), and 3) subglottal pressure level ($P_s$). These laryngeal mobility space trajectories are then converted into sensory task space trajectories consisting of two auditory dimensions (vocal $f_o$ and *SPL*). With this inclusion, both laryngeal and vocal tract articulatory components of the speech sound are controlled via the feedforward-feedback control architecture of LaDIVA. The extended BCM model is shown in *Fig 2*. A more detailed technical description of LaDIVA is provided in *Materials and Methods*. Note that the current implementation of LaDIVA does not contain a somatosensory task space representation for laryngeal mobility space variations due to the limited somatosensory task space representation in DIVA for laryngeal motor control.

## Results

Here we present an evaluation of LaDIVA's ability to simulate human laryngeal motor control. A series of model simulations were carried out to validate the model qualitatively and quantitatively.

### Model response to reflexive and adaptive auditory perturbations

We used data collected from a previous behavioral experiment [68] to validate LaDIVA. A series of model simulations replicating vocal $f_o$ reflexive and adaptive perturbation paradigms were conducted. LaDIVA simulation responses were comparable with behavioral responses to vocal $f_o$ reflexive and vocal $f_o$ adaptive perturbations of auditory feedback of laryngeal features (i.e., vocal $f_o$) collected in adults with typical speech [68]. *Fig 3* shows the simulation responses

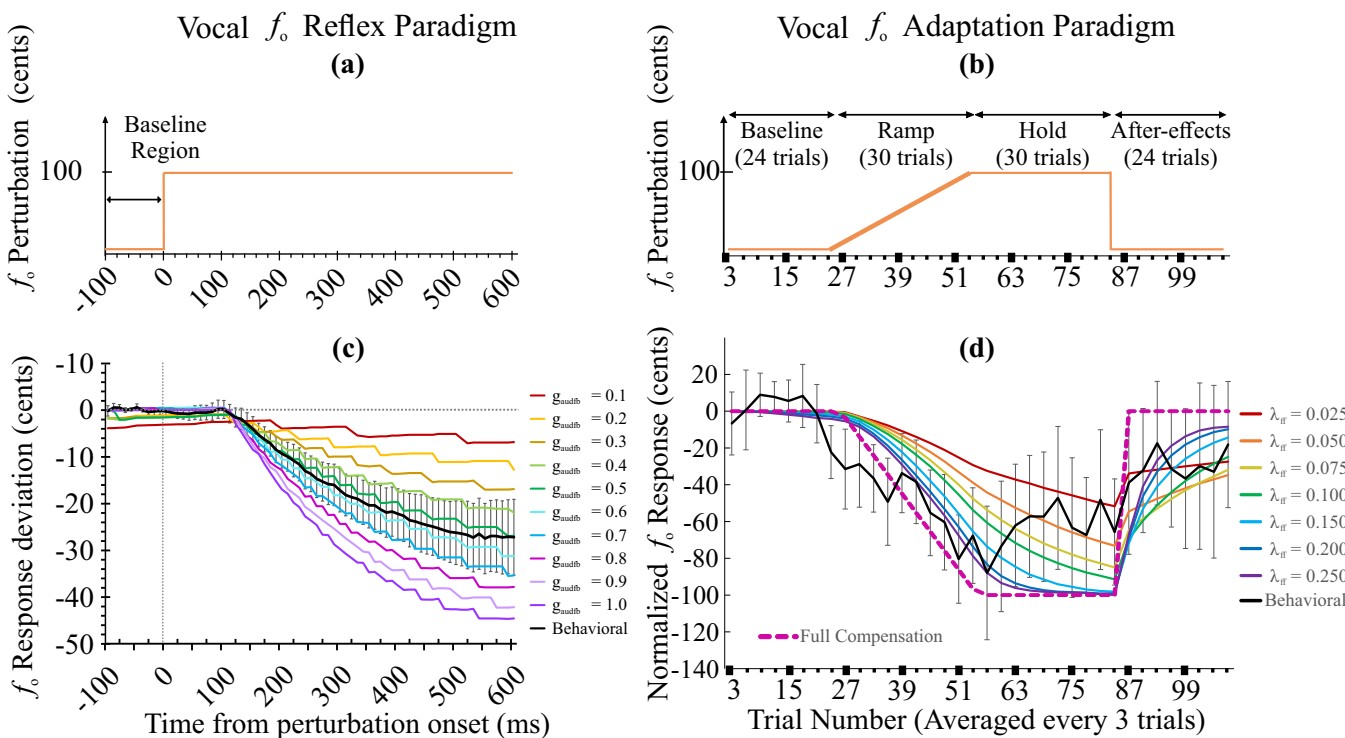

**Fig 3. Simulation responses of LaDIVA compared with the mean group response (20 adults with typical speech) of the behavioral dataset.** (a) Applied $f_o$ perturbation for a perturbed trial in the vocal $f_o$ reflexive paradigm. (b) Vocal $f_o$ perturbation across all 108 trials of the vocal $f_o$ adaptive paradigm. (c) Simulation response for vocal $f_o$ reflexive paradigm. (d) Simulation response for vocal $f_o$ adaptation paradigm. All simulations in panel (c) and (d) were conducted under initial laryngeal muscle activation settings (Case B: $a_{CT} = 0.169$, $a_{TA}$ $a_{TA} = 0.175$, and $P_s = 800$ Pa). Group mean response of behavioral dataset in black with 95% CI error bars).

in LaDIVA in comparison with the mean group response (20 young adults with typical speech) of the behavioral dataset.

In LaDIVA we focused on five tunable parameters for the simulations: two parameters from DIVA and three from the extended BCM. These were defined within physiologically relevant ranges and tuned to best match behavioral data from the previous study [68]. The DIVA parameters are auditory feedback gain ($g_{audfb}$), and feedforward learning rate ($\lambda_{ff}$). See *Materials and Methods* for parameter incorporation via mathematical equations. The extended BCM parameters are the mobility space targets for CT muscle activation ($a_{CT}$), TA muscle activation ($a_{TA}$), and subglottal pressure ($P_s$). All other parameters in DIVA and extended BCM were set to replicate male VFs as described in Story and Titze Body-Cover Models [46,48], with a baseline vocal $f_o$ of 134 Hz [79] and a vocal intensity represented by a radiated sound pressure level of 75 dB SPL at 30 cm from the lips.

The auditory feedback gain ($g_{audfb}$) was determined using a reflexive paradigm simulation. The vocal $f_o$ of the auditory feedback signal fed into the model was artificially shifted by +100 cents (1 semitone) after 500 ms from vocal onset (see *Fig 3A*: only 100 ms before perturbation onset is shown) and the perturbation was sustained until the end of the utterance. LaDIVA was used to replicate the real-time response for the utterance of a target vowel /ɑ/, with the added auditory perturbation. The LaDIVA auditory state indicates the current auditory feedback the model receives, which is comparable to the headphone signal a participant hears in an experimental session. See Eq 10B in *Materials and Methods* for details. Thus, the system response to the auditory perturbation can be calculated as the difference between the current

**Table 2. Goodness-of-fit statistics for behavioral dataset for vocal $f_o$ reflexive paradigm explained by model simulations.** AUC = area under the curve.

| Feedback Gain | 0.1 | 0.2 | 0.3 | 0.4 | 0.5 | 0.6 | 0.7 | 0.8 | 0.9 | 1 |
|---|---|---|---|---|---|---|---|---|---|---|
| R-squared | -0.40 | 0.19 | 0.62 | 0.87 | **0.98** | 0.97 | 0.83 | 0.59 | 0.24 | -0.17 |
| RMSE | 12.67 | 9.63 | 6.61 | 3.90 | **1.49** | 1.95 | 4.41 | 6.87 | 9.32 | 11.62 |
| Absolute Difference AUC | 596.06 | 473.42 | 312.46 | 161.15 | **15.79** | 93.14 | 235.85 | 375.19 | 513.57 | 645.51 |

auditory state and the applied auditory perturbation. Normalized vocal $f_o$ responses for vocal $f_o$ reflexive paradigm simulations are provided in *Fig 3C* for varied auditory feedback gains ($g_{audfb}$ ranging from 0.1–1.0). The model simulation results were overlaid with the behavioral data of the reflexive group mean $f_o$ response, and the best fit for feedback gain was selected. With the lowest RMSE and difference between area under the curves, $g_{audfb} = 0.5$ provided the best fit for the behavioral mean vocal $f_o$ reflexive response (RMSE = 0.91, $R^2$ = 1.0; See *Table 2*). Note that the feedforward learning rate parameter ($\lambda_{ff}$) was kept constant for the auditory feedback gain parameter exploration as reflexive paradigms are designed to avoid learning effects by avoiding the presentation of sustained and predictable perturbations to auditory feedback that may cause persistent auditory feedback errors.

The feedforward learning rate parameter ($\lambda_{ff}$) was determined through the use of a vocal $f_o$ adaptive paradigm simulation ($g_{audfb}$ was kept constant). The vocal $f_o$ of the auditory feedback signal fed into the model was perturbed according to four ordered phases; 1) Baseline: no perturbations, 2) Ramp: vocal $f_o$ shift increasing by +3.3. cents in each adjacent trial, 3) Hold: vocal $f_o$ shifted by +100 cents, and 4) Aftereffect: no perturbation applied. See *Material and Methods*: *Behavioral study dataset* section for more details. For perturbed trials, the vocal $f_o$ of the auditory feedback signal fed into the model was shifted before vocal utterance onset and the perturbation was sustained until the end of the utterance in the simulation. The model simulation was run for 108 adjacent simulations (i.e., trials) to replicate 108 trials of target vowel /α/ utterances, with added auditory perturbation in specified phases. The vocal response is a combination of the feedforward and feedback motor commands acting on current vocal production. See *Material and Methods* Eq 9 for more details. In order to extract the acoustic response driven by feedforward motor commands alone, an analysis window from 40 ms to 120 ms after the vocal onset was selected. This analysis window was selected to target the feedforward motor command based response as auditory feedback is assumed to affect current vocal productions with a latency of 120 ms [63]. The initial 40 ms of the response signal was discarded to avoid vocal onset variability generally present in human vocal production. The mean response $f_o$ for the analysis window of each trial was calculated to represent the adaptive response per each trial. Note that in the model simulations, the analysis window captures pure feedforward signals as the modeled auditory feedback signal latency is 120 ms. Although this analysis window is identical to the behavioral data analysis, in behavioral data from human speech motor control systems, there could be somatosensory feedback signals (i.e., with feedback latency 65–75 ms; [80]) present and affecting the speech signal in the selected analysis window. Normalized vocal $f_o$ responses per each trial for the adaptive paradigm simulations are provided in *Fig 3D* for the preselected auditory feedback gain via vocal $f_o$ reflexive

**Table 3. Goodness-of-fit statistics for behavioral dataset for vocal $f_o$ adaptive paradigm explained by model simulations.** AUC = area under the curve.

| Feedforward learning rate | 0.025 | 0.050 | 0.075 | 0.100 | 0.150 | 0.200 | 0.250 |
|---|---|---|---|---|---|---|---|
| R-squared | 0.16 | 0.33 | 0.33 | **0.32** | 0.26 | 0.21 | 0.20 |
| RMSE | 24.02 | 21.44 | 21.43 | **21.57** | 22.55 | 23.24 | 23.41 |
| Absolute Difference AUC | 545.85 | 227.93 | 52.90 | **27.58** | 131.07 | 184.41 | 203.94 |

simulations ($g_{audfb}$ = 0.5) and for different feedforward learning rates ($\lambda_{ff}$ ranging from 0–0.25). Although an ideal fit was not observed, with the lowest RMSE and difference between area under the curves, $\lambda_{ff}$ = 0.1 provided a consistent fit for the behavioral mean vocal $f_o$ adaptive response (RMSE = 21.57, $R^2$ = 0.32; See Table 3).

## Motor equivalence in simulated vocal productions

The initial biomechanical conditions of the VFs define how the compensatory responses to auditory perturbations are handled in dimensions that are acoustically less important. This is known as *motor equivalence* [20]. Based on this theory, we carried out two sets of simulations to characterize LaDIVA's behavior. Initial laryngeal mobility space variables $a_{CT}$, $a_{TA}$, and $P_s$ were preset parameters for the extended BCM. We used these parameters to characterize the model behavior in a controlled manner to understand how differences in CT and TA muscle activations affect the acoustic outcomes of the system. We confirmed that a variety of combinations of $a_{CT}$ and $a_{TA}$ parameters could produce the same vocal $f_o$ while subglottal pressure is kept constant (i.e., $P_s$ = 800 Pa). See *Fig 4A* Muscle Activation Map for vocal $f_o$ ($f_o$ = 134 Hz contour line). Vocal SPL also remained fairly constant during these simulations (range: 71–76 dB SPL), which is consistent with *in vivo* work that observed minimal change in SPL in the lower register (i.e., including modal phonation) when $P_s$ is kept constant [81]. See *Fig 4B* Muscle Activation Map for vocal SPL. Four combinations of initial $a_{CT}$ and $a_{TA}$ levels were used to simulate four different initial conditions in phonation (see *Table 4* and *Fig 4A and 4B* cases). For each of these cases, a simulation of a vocal $f_o$ reflexive paradigm was carried out and the model outputs of each paradigm were compared across cases (see *Fig 4*). We observed that, although $a_{CT}$ and $a_{TA}$ trajectories varied across the simulations (*Fig 4C and 4D*), the vocal $f_o$ responses (i.e., the acoustic output) were comparable to the behavioral dataset across cases (*Fig 4E*). For very small values of $a_{CT}$, we see $a_{TA}$ dominance in $f_o$ control (shown in case A in Fig 4). For the rest of the cases, $a_{CT}$ takes precedence in controlling vocal $f_o$ and $a_{TA}$ plays a less dominant role. This is consistent with prior literature that suggests antagonistic relationships between vocal $f_o$ and $a_{CT}$ and $a_{TA}$ across the chest register [55,82,83]. The vocal SPL target range was maintained at 68–80 dB for the simulations to mimic the vocal SPL outputs expected in the behavioral study (i.e., participants were instructed to speak in their typical speaking voice). As we maintained a constant subglottal pressure through all simulations, in the range of $a_{CT}$ and $a_{TA}$ considered, the change in vocal SPL output was less than 1 dB SPL. Thus, the vocal SPL outputs did not exceed the target range, and no corrective feedback signals modifying SPL output were generated. As a result, no deviations in vocal SPL were observed for the simulations. For all simulations in Cases A through D, the auditory feedback gain and feedforward learning rate were kept constant (i.e., $g_{audfb}$ = 0.5 and $g_{ff}$ = 0.1).

## Prosodic contour simulations

Auditory perturbations have been carried out to understand the vocal motor control mechanisms for control of vocal $f_o$ and SPL in a task-dependent manner during the production of suprasegmental features (e.g., stress, intonation) in running speech [84–86]. Different prosodic contours are generated by participants for running speech utterances and vocal $f_o$ perturbations are carried out at different stages of the utterance to investigate the task-relevance of compensatory responses to these perturbations. To model these dynamic perturbations, as an initial step, speech production models should be able to generate running speech utterances with different vocal $f_o$ contours. So far, the DIVA model has been unable to simulate dynamic vocal $f_o$ perturbations as vocal $f_o$ and SPL were not controlled variables in the model. As an initial step to assess and showcase the LaDIVA model's ability to conduct dynamic vocal $f_o$

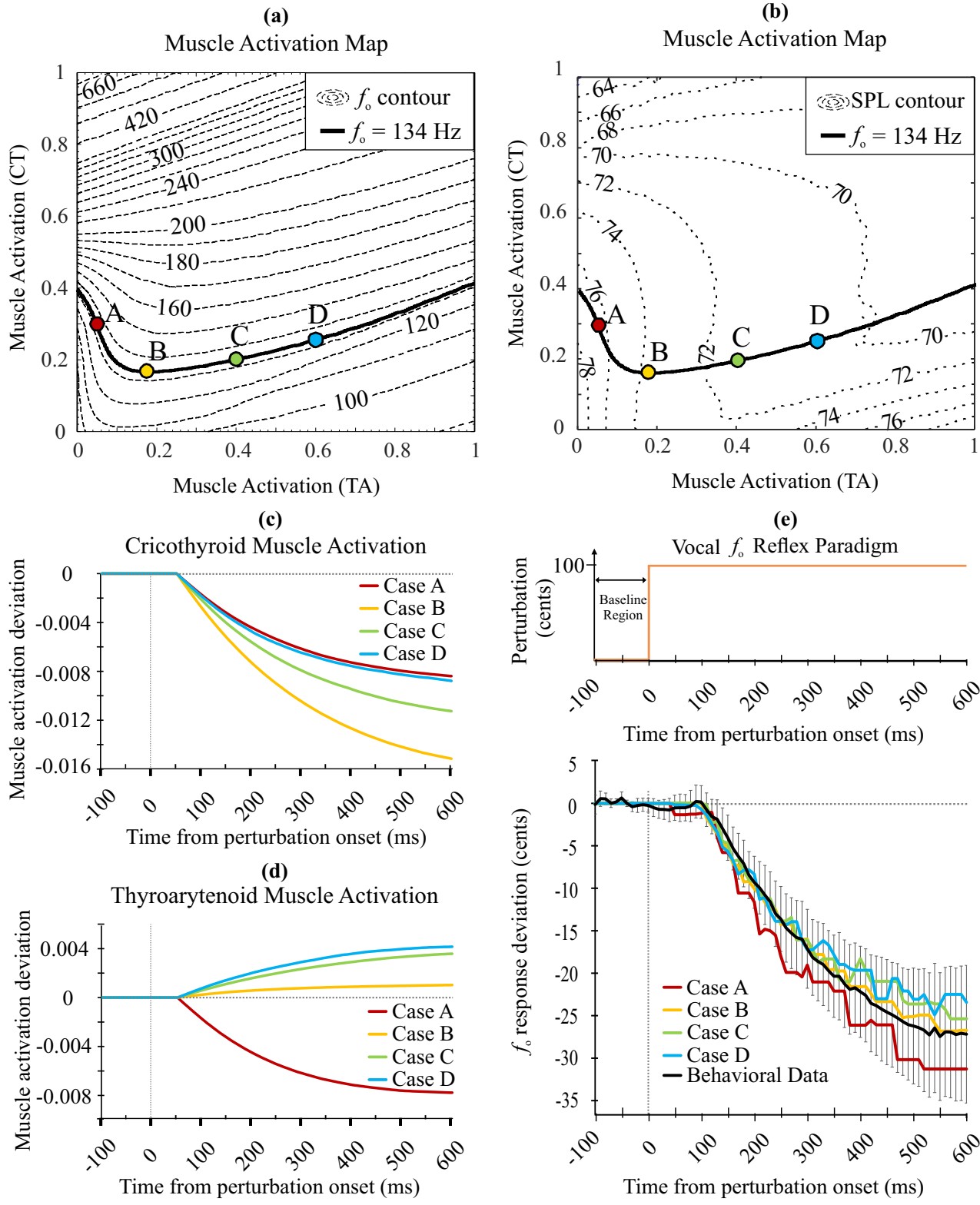

**Fig 4. Simulation responses of LaDIVA for four different initial laryngeal muscle activation levels.** (a-b) Cases A, B, C, and D muscle activation levels are indicated by dots on the $f_o$ = 134 Hz solid line, with subglottal pressure (P$_s$) = 800 Pa. (a) shows $f_o$ contours (Hz units) and (b) shows SPL

contours (dB units). (c) Cricothyroid muscle activation level trajectory over the simulation for all initial conditions. (d) Thyroarytenoid muscle activation level trajectory over the simulation for all initial conditions. (e) Vocal $f_o$ simulation response for vocal $f_o$ reflexive paradigm in comparison with mean group response (20 adults with typical speech) of behavioral dataset marked in black with 95% CI error bars. *Note*: vocal SPL output deviation was zero as the SPL target range was set to 68–80 dB and all cases remained within target range.

perturbations, we carried out a series of model simulations incorporating four intonation patterns to a single base statement. These examples showcase that the LaDIVA model can be used to embed different prosodic contours in phrases as vocal $f_o$ is a controlled component. We expect that this methodology can be used in the future to generate stimuli for auditory vocal $f_o$ perturbations of running speech.

LaDIVA contains an acoustic synthesizer that is adapted from DIVA and modified to accommodate the extended BCM. See *Materials and Methods* for more information on the acoustic synthesizer. The prosodic contour simulations can also be used to assess the function of the acoustic synthesizer and the naturalness of the acoustic outputs generated. See *S1 Audio* for audio recordings of original and LaDIVA-synthesized utterances. The statement "Buy Bobby a Puppy" was recorded from one of the authors (F.H.G.) with four different prosodic contours, providing different intonation patterns, and thus, different contextual information. The sentence was also synthesized in the model by defining auditory and somatosensory targets in the articulatory domain such that a monotonic production of the sentence was produced. The simulated sentence duration was 1815 ms. Original recordings were time-normalized by warping each recording in the time domain to match the model production. From each time-warped recording, the pitch contour was extracted. Initial $a_{CT}$ and $a_{TA}$ levels of case B were used to conduct all pitch contour simulations, resulting in an initial vocal $f_o$ of 134 Hz. A baseline vocal $f_o$ increase of 30 Hz was applied to all extracted pitch contours to match the original recordings (i.e., of a male with 100 Hz vocal $f_o$) to model production that has a baseline $f_o$ of 134 Hz. The pitch contours were then smoothed using a 5-point moving-average, before being fed to the model as vocal $f_o$ target contours. A series of simulations was carried out until the output of each vocal $f_o$ pitch contour converged to the target contour. The number of iterations required for contour convergence was determined be faster or slower based on the model parameters used. The model parameters were set as following for the simulations shown in *Fig 5* ($g_{fb\_aud} = 1$; $\lambda_{ff} = 0.5$). See *Fig 5* for time-warped versions of the original pitch contours of the four different intonations patterns plotted against synthesized versions of the same contours from LaDIVA. These results showcase LaDIVA's potential to perform simulations for studies incorporating intonation, tonal languages, and running speech $f_o$ perturbations [85, 86].

## Discussion

In the current study, we introduced LaDIVA, a neurocomputational model of laryngeal motor control. LaDIVA extends the well-established neurocomputational model DIVA,

**Table 4. Initial laryngeal intrinsic muscle activation levels applied for simulations.** $a_{LC}$ was kept fixed at 0.5 for all simulations.

|  | Case A | Case B | Case C | Case D |
|---|---|---|---|---|
| **$a_{CT}$ and $a_{TA}$ relationship** | $a_{CT} > a_{TA}$ | $a_{CT} = a_{TA}$ | $a_{CT} < a_{TA}$ | $a_{CT} \ll a_{TA}$ |
| **$a_{CT}$ level** | 0.300 | 0.169 | 0.202 | 0.256 |
| **$a_{TA}$ level** | 0.050 | 0.175 | 0.400 | 0.600 |
| **Subglottal Pressure ($P_s$)** | 800 | 800 | 800 | 800 |
| **Vocal $f_o$ output (Hz)** | 134.20 | 133.98 | 133.96 | 133.97 |
| **Sound Pressure Level (dB SPL)** | 76 | 74 | 71 | 71 |

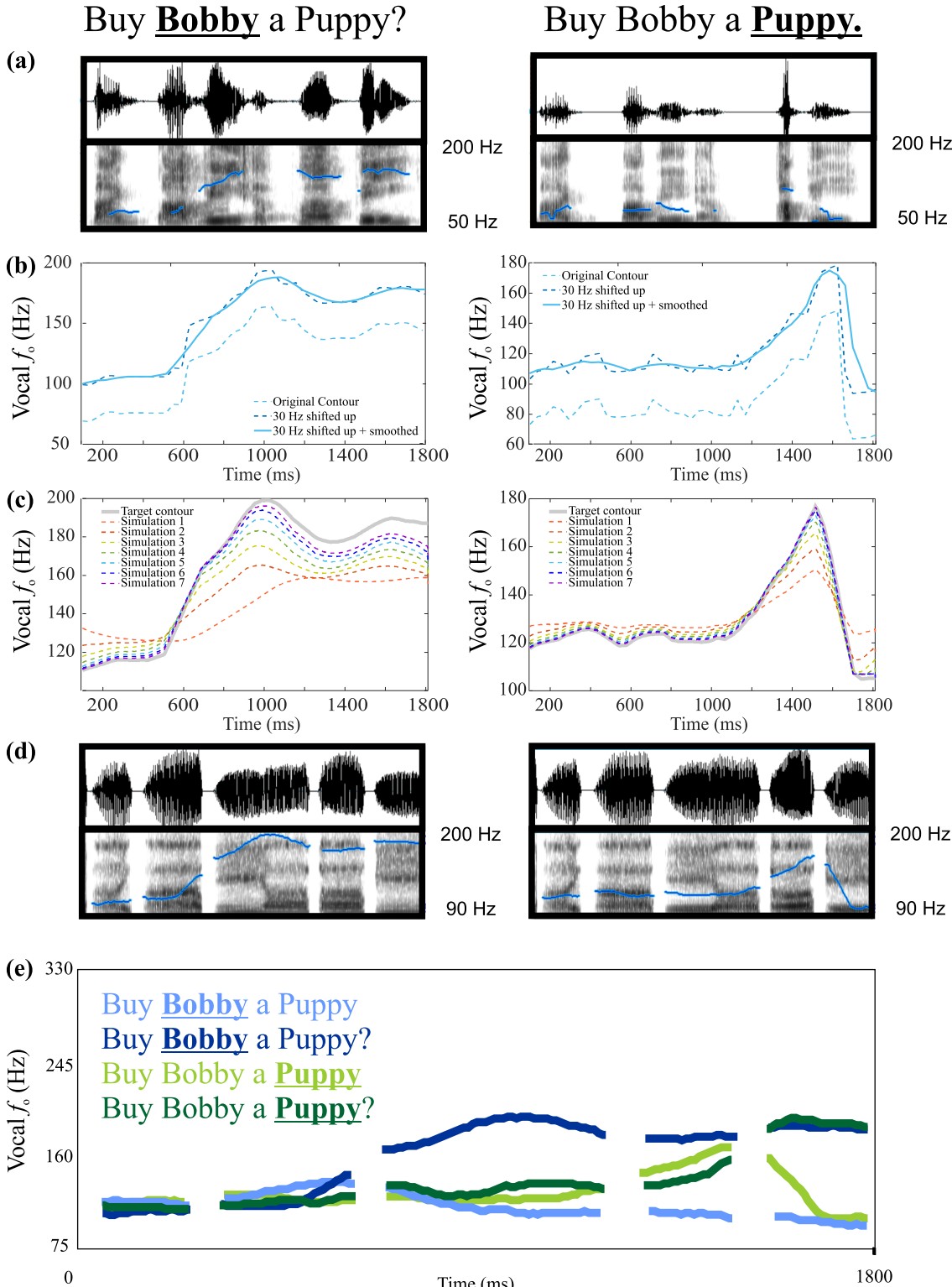

**Fig 5. Prosodic Contours of "Buy Bobby a Puppy".** Fig 5A–5D Vocal $f_o$ contour extraction, transformation, model simulation, and model synthesis output shown for two intonation patterns. (a) Original recordings recorded from author F.H.G. (male 100 Hz vocal $f_o$). (b) A baseline vocal $f_o$ increase of 30 Hz was applied to all extracted pitch contours to match case B model mean vocal $f_o$ of 134 Hz. The pitch contours were smoothed using a 5-point moving-average. (c) Transformed pitch contours were then fed as vocal $f_o$ target contours to the model and a sequence of simulations were carried out until the output vocal $f_o$ contour matched the target contour.

(d) output signal and vocal $f_o$ contour produced by LaDIVA acoustic synthesizer. (e) Simulated vocal $f_o$ contours of four different intonation patterns of producing the sentence 'Buy Bobby a Puppy' time warped to be the same duration in time. Bolded and underlined sections highlight stressed words. *Note*: *model parameters were set as following for these simulations (learning rate = 0.5; feedback gain = 1). The contour convergence process can be faster or slower based on the parameters used.*

incorporating a biomechanical VF model for dynamic adjustment of laryngeal function. Model performance was validated via empirical evidence from a behavioral study on adults with typical speech involving reflexive and adaptive vocal $f_o$ auditory feedback perturbation paradigms. Through the application of LaDIVA, we were able to successfully replicate both the reflexive and adaptive auditory vocal $f_o$ responses observed in the behavioral data. The simulations also provided CT and TA laryngeal muscle activation levels across time for the paradigms. However, these predicted muscle activations remain to be experimentally validated via behavioral studies. Intrinsic laryngeal muscle activity is difficult to obtain in behavioral settings due to its invasive recording procedures using laryngeal electromyography techniques [34,35,37]. Only one previous study measured laryngeal electromyography responses to a vocal $f_o$ auditory feedback reflexive perturbation paradigm [36]. The paradigms conducted in that study with conversational speech did not elicit observable patterns in CT and TA muscle activation levels and the authors speculated that the muscle activation levels were not detectable due to high levels of underlying neuromuscular signal noise. Nonetheless, the current model results are supported by a prior study conducted using graded muscle simulations on an in *vivo* canine larynx that observed general laryngeal motor equivalence, i.e., the ability of the larynx to produce same vocal $f_o$ and SPL outputs with multiple combinations of intrinsic laryngeal muscle activations [81]. Thus, modeling efforts could be an effective way to investigate intrinsic laryngeal muscle activity related to vocal $f_o$ reflexive paradigms further in future.

Auditory feedback gain and feedforward learning rate in LaDIVA are model parameters that can be modified to represent neural motor control impairments related to neurological (i.e., Parkinson's disease; PD) or functional voice disorders (i.e., vocal hyperfunction). Hyperactive auditory feedback responses in persons with PD who are off medication have been noted by multiple altered auditory feedback behavioral studies for vocal $f_o$ [72–75] and for vocal SPL [73]. Reduced adaptive responses were also observed in several studies carried out with persons with PD (vocal $f_o$: [69], vocal SPL: [87]). Overall, these studies suggest that persons with PD have hyperactive auditory feedback and weakened auditory-motor integration that causes impaired feedforward motor control. Thus, the vocal motor control deficits in PD can be modeled using LaDIVA by increasing the auditory feedback gain and decreasing the feedforward learning rate. Altered auditory feedback behavioral studies have also indicated that there is a possible auditory-motor phenotype for individuals with vocal hyperfunction in which atypical vocal motor control is observed [57,70,71]. Deficits in auditory motor integration has been identified via adaptive vocal $f_o$ studies conducted in individuals with vocal hyperfunction [57,70]. Thus, the suspected vocal motor control deficits in vocal hyperfunction can be modeled using LaDIVA by decreasing the feedforward learning rate. However, vocal hyperfunction has not been sufficiently investigated to conclude if the origins are neurological, anatomical, or has components from both neural control and biomechanical control. The objective of LaDIVA was to combine a neurocomputational model of vocal motor control and a biomechanical model of the peripheral laryngeal apparatus, thus providing a comprehensive model of voice motor control. We can manipulate the model's neurocomputational as well as biomechanical parameters to investigate and gather insight on how the respective changes affects the vocal motor outcomes of the system.

An interesting finding of the current study was that LaDIVA yielded similar acoustic outputs to a reflexive paradigm across different initial muscle activation conditions. Based on VF physiology, laryngeal muscle activations vary across individuals with typical laryngeal function based on sex, age, and different voicing conditions [88]. These observations might be instrumental in understanding how similar acoustic outputs are achieved despite individual variability in internal neuromuscular activations. LaDIVA can be used with individualized parameters to reflect physiological changes in VFs due to age or sex, and the variability in laryngeal muscle activations can be quantified, using simulation outcomes such as vocal $f_o$ reflexive responses. These quantitative results may be instrumental in identifying what degree of variability is due to variations in laryngeal physiology. At the same time, these results caution us against using only the acoustic vocal $f_o$ response measures in characterizing neural laryngeal motor control function, specifically in individuals with voice disorders characterized by increased laryngeal muscle tension. For example, individuals with vocal hyperfunction are thought to exhibit higher laryngeal muscle activation [89,90]. However, a recent study found no significant differences in vocal $f_o$ reflexive responses in individuals with vocal hyperfunction compared to individuals without vocal hyperfunction [57]. Thus, individuals with vocal hyperfunction seem to be generating similar acoustic outcomes as individuals without vocal hyperfunction, but presumably with higher intrinsic muscle activation levels. This could be a possible explanation for the clinical observations of low vocal efficiency in individuals with vocal hyperfunction [90,91].

We note that LaDIVA's adaptive paradigm simulations are not as well-fit to the behavioral data as reflexive paradigm simulations. There could be multiple reasons for this observation, including: 1) vocal motor variability not being a modeled characteristic in LaDIVA, 2) somatosensory feedback not being a modeled component, 3) a single parameter (i.e., feedforward learning rate) defining adaptive behavior in LaDIVA, and 4) identical feedback error contributions being used for feedforward and feedback motor control. Trial-to-trial noise in speech acquisition, production, and learning are inherent characteristics in empirical data related to speech motor control [92–94]. However, the current implementation of LaDIVA contains no model characteristic defining vocal motor production or perception variability. Behavioral responses used in the current study clearly show the trial-to-trial noise variations whereas the model simulations are noise-free and smooth across trials. See *S2 Text* for production noise related variations observed in simulated adaptive paradigm outputs when varying levels of production noise are included in the LaDIVA model. Adding noise-based variability at production, perception, and/or learning levels will be carried out in future iterations as it requires qualitative and quantitative specification of noise characteristics via behavioral paradigms. Somatosensory feedback of the model remains unchanged during auditory feedback perturbations and would cause the feedforward control system to correct for accumulated somatosensory errors over time. Thus, the overall effect would yield a lower $f_o$ magnitude for the simulated adaptive response. In addition, identical feedforward learning rates are chosen for the incorporation of auditory and somatosensory feedback to feedforward controller in LaDIVA. As somatosensory feedback contributions are set to zero in the current iteration of LaDIVA, this assumption does not affect the overall adaptive response simulations. Furthermore, a recent study examined the error sensitivity of feedforward and feedback motor control systems using adaptive paradigms in first and second vowel formants and the state-space model [95]. The study provided initial evidence that feedforward and feedback control systems may function independently, with higher error sensitivity in feedback control compared to feedforward control. These recent findings are in contrast to the DIVA model assumption that the feedback and feedforward motor controllers share identical error sensitivity. LaDIVA

shares this assumption, which could be a possible explanation for poorer simulation fits for the adaptive paradigm compared to reflexive paradigm.

In this validation study, we have demonstrated LaDIVA's capability to simulate both adaptive and reflexive paradigms of sustained vowels as well as its capability to handle running speech utterances (see *Results* section on prosodic contours). LaDIVA provides the flexibility to observe feedback- and feedforward-based $f_o$ contours across the time span of a trial, as well as to focus on specific time windows. Thus, LaDIVA can be used to replicate and study characteristics of feedback and feedforward responses separately (i.e., via late and early time windows in each trial) using adaptive paradigm simulations [95]. The capability to simulate running speech utterances embedded with laryngeal motor control gives LaDIVA the unique capability to model vocal $f_o$ perturbation studies in running speech, which has so far not been modeled to the best of our knowledge. Thus, LaDIVA may allow for the generation of better hypotheses for behavioral studies related to vocal $f_o$ perturbations in running speech. We have carried out parameter fitting for auditory feedback gain and feedforward learning rate parameters in the current study as a validation of the model's capability to model instances of laryngeal motor control with hyperactive auditory motor control mechanisms and weak auditory-motor integration capabilities. Thereby, we showcase LaDIVA's capability to replicate vocal motor control deficits that have been documented in voice disorders related to PD and vocal hyperfunction. Thus, the parametric fitting carried out in the validation of LaDIVA serves a larger purpose than merely showcasing that the model can fit data. It sheds light to the neurobiological control of vocal $f_o$.

## Comparisons with simpleDIVA

SimpleDIVA is a 3-parameter model recently developed to fit average data from different cohorts under different auditory and somatosensory adaptive paradigms. It can be used to predict auditory and somatosensory feedback gains, as well as the feedforward learning rates used by each behavioral dataset [76]. SimpleDIVA can be used to predict the outcome $f_o$ contours of specific adaptive auditory feedback paradigms based on prior assumptions about feedback gains and the feedforward learning rate. SimpleDIVA is a powerful modeling tool due to its simplicity leading to unique solutions with limited degrees of freedom. Although simpleDIVA successfully abstracts the DIVA architecture to model both articulatory and laryngeal motor control, its simplification of DIVA to three mathematical equations has its own limitations.

SimpleDIVA and LaDIVA are radically different in both their architecture and more importantly, their intent. LaDIVA is a control model of vocal and supralaryngal production that is capable of learning. Moreover, LaDIVA is capable of simulating acoustic responses and brain activations to behavioral paradigms. SimpleDIVA is, conversely, a way to fit a set of equations to time-series data to estimate gain parameters. Thus, LaDIVA can complement simpleDIVA in studying laryngeal motor control. LaDIVA can be used to support the development of hypotheses that can be tested empirically in behavioral studies to ultimately advance the field of laryngeal motor control. LaDIVA increases the predictive power in hypotheses by incorporating biomechanical and physiological restrictions of the VFs in producing vocalizations. A biologically relevant neural model of laryngeal motor control provides a unique platform to model voice disorders that are hypothesized to have neurological origins.

The availability of multiple degrees of freedom in LaDIVA can be a disadvantage due to the large number of non-unique solutions the data fits could provide. However, the inclusion of the VF model provides its own physiologically valid restrictions on laryngeal motor control function of LaDIVA and thus safeguards LaDIVA from resulting in biologically infeasible

solutions. Moreover, due to the large number of defined biomechanical control parameters, LaDIVA possesses the ability to carry out subject-specific simulations. This feature is inherited to LaDIVA from the extended BCM, which has been successfully used to conduct subject-specific modeling in individuals with vocal hyperfunction [14,96]. Although simulating subject-specific data is out of scope for the current investigation, this is a potential area of future expansion of LaDIVA.

## Auditory acuity and auditory target size

Vocal $f_o$ acuity can be assessed via the minimum noticeable difference in vocal $f_o$, commonly referred to as the Just Noticeable Difference (JND; inversely related to vocal $f_o$ acuity) between two frequencies in behavioral paradigms. In LaDIVA, vocal $f_o$ acuity is a preset parameter. See *Materials and Methods*: *targets* section for more detail. The JND for vocal $f_o$ in LaDIVA was set during all simulations to be significantly smaller compared to the maximum perturbation magnitude utilized in the perturbation paradigms (here, vocal $f_o$ JND = 5 cents $<<$ maximum perturbation magnitude = 100 cents). Here, a five-cent JND threshold was selected to reflect a model with excellent $f_o$ acuity, since prior studies have indicated that participants respond to $f_o$ shifts as small as 10 cents in reflexive paradigms [97,98]. However, vocal $f_o$ acuity has been previously shown to differ across individuals and thus could be varied as a parameter in future versions of LaDIVA [99–101]. Prior studies have provided evidence that participants with better vowel formant acuity generate larger compensatory response magnitudes to adaptive vowel formant perturbation paradigms [102–104]. A single study using DIVA was able to replicate this relationship for simulated participant responses, with vowel formant target sizes modeled using Gaussian distributions with varying distribution variances (i.e., vowel formant acuity was inversely related to distribution variance; [104]). However, in terms of laryngeal motor control, there have been conflicting prior study observations about the relationship between auditory feedback perturbation magnitudes and corresponding vocal $f_o$ acuity, suggesting that there could be a complex relationship (or lack thereof) between generating corrective feedback for auditory feedback errors and perceiving those feedback errors [68,98,105,106]. Including vocal $f_o$ acuity as a variable parameter will allow for modeling intra-speaker $f_o$ acuity variability in LaDIVA, helping to interpret these recent findings.

## Limitations

The current implementation does not include physical tissue-aero-acoustic interactions at the glottis (i.e., the modification of VF dynamics due to acoustic and aerodynamic feedback; [81]). The DIVA model provides a modified Maeda model [78] based vocal tract, which is integrated in LaDIVA. When coupling effects due to vocal tract acoustics are neglected, the laryngeal adjustments resulting from the coordinated control of VF stiffness, geometry, and position do not adequately model the effects on the source intensity or vocal quality. This is backed by laryngeal, physical, or computational models of phonation [107]. The effect of laryngeal adjustments on vocal intensity becomes more relevant in the presence of an interactive vocal tract model [14]. DIVA currently contains an articulatory synthesizer that incorporates simplified, custom source-filter interactions that are mathematically defined. However, a more physiologically relevant source-filter interaction can be modeled using biomechanical VF models with vocal tract interactions. The Maeda model used for vocal tract modeling in DIVA is compatible with any biomechanical VF model. Moreover, an added advantage specific to Maeda's model is that it allows simulation of vocal tract elongation/shortening, which is perceptually important for speech simulation [78,108,109]. Thus, a possible future modification that can be implemented in LaDIVA is to incorporate the tissue-aero-acoustic interaction at the glottis

using the extended BCM, and the acoustic effects due to subglottal tract dynamics by interconnecting the BCM with the Maeda model, making LaDIVA to be more physiologically relevant.

Currently, LaDIVA focuses mainly on the auditory feedback controller and auditory feedback perturbations. We kept the somatosensory feedback controller offline for the current version by setting the somatosensory feedback gain to zero. However, laryngeal motor control is reliant on feedforward motor commands and sensory feedback provided via both auditory and somatosensory feedback subsystems [110,111]. Thus, the contributions of the somatosensory feedback controller cannot be ignored completely. Many auditory sensorimotor adaptation studies in vowel formants as well as vocal $f_o$ showcase that the amount of adaptation plateaus well before reaching full compensation [66,68,69,104,112]. This partial compensation is explained in the DIVA model as being driven by the competition between counteracting auditory and somatosensory feedback signals. At the beginning of a trial when auditory feedback is artificially modified, the somatosensory feedback controller does not find any discrepancies between somatosensory feedback received (unaltered) and desired somatosensory targets. As the system starts compensating for the artificial auditory feedback alterations by modifying subsequent feedforward motor commands and producing altered speech sounds, the somatosensory feedback also starts to be altered, causing errors between somatosensory feedback and desired somatosensory targets. These counteracting feedback signals eventually cause the compensatory response to plateau, before it fully compensates for the altered auditory feedback. This reasoning may explain why LaDIVA's adaptive simulations do not produce $f_o$ contours that fit empirical data as well as its reflexive simulations.

With regard to reflexive vocal $f_o$ paradigms (i.e., where real-time feedback error correction is investigated) there are also behavioral observations of partial compensation [113–117]. These compensatory magnitudes are explained in the DIVA model as being driven by the competition between feedforward and feedback control systems, as well as the competition between auditory and somatosensory feedback mechanisms. Since the corrective auditory feedback-driven motor command is (weighted and) summed with the feedforward command to produce the final speech signal, theoretically the system can maximally produce a partial compensation of 50% (see Eq 9). This is a strong prediction of the model and can be observed via the reflexive paradigm simulations (see *Fig 3C* and *Fig A in S1 Text*). In these simulations, setting the total feedforward gain to 1, the auditory feedback gain to 1, and the somatosensory feedback gain to 0 results in a partial compensation of 50 cents for a 100 cent- reflexive perturbation of auditory feedback, which converts to a 50% (partial) compensation. With non-zero somatosensory feedback contributions, the partial compensation will be even less, as observed in literature [113–117].

In LaDIVA, the glottal constriction is the only parameter for somatosensory representation of vocal motor function and it was not controlled by the extended BCM in the current version (i.e., it was predefined as constant values for voiced and unvoiced portions of productions and not dependent on LCA or PCA muscle activity in the extended BCM). Better understanding of the somatosensory feedback controller contribution to laryngeal motor control needs to be obtained to define accurate somatosensory representations for vocal features. For example, it remains unknown whether vocal fold collision and tactile feedback during voicing onset and offsets are monitored via the same somatosensory feedback mechanisms as those that monitor vocal $f_o$ control via VF vibratory activity during sustained phonation. Thus, more information about the types of somatosensory feedback (i.e., tactile, vibratory, and proprioceptive) used in laryngeal motor control needs to be acquired to define multiple somatosensory representations for laryngeal motor control in LaDIVA. Prior research suggests that somatosensory feedback provides prephonatory information about the intrinsic laryngeal musculature and the length of the VF (i.e., proprioceptive feedback), prior to the onset of the vocalization [118,119]. This

may indicate that auditory and somatosensory feedback controllers operate over different time scales when providing corrective signals for laryngeal motor control. Finally, individual variability in the sensitivity to or reliance on somatosensory feedback may be important. Women have shown exhibited increased sensitivity to somatosensory feedback manipulations relative to men, suggesting a higher density of mechanoreceptors in women compared to men due to anatomical differences [120]. Furthermore, prior work using vowel formants suggests that there is a sensory preference in the articulatory domain for either auditory and somatosensory feedback that varies across individuals [121,122]. In sum, there are many factors affecting the contribution of somatosensory feedback to laryngeal motor control, which should be incorporated into future modeling efforts.

When repeated vocal productions are carried out (e.g., via adaptive paradigms), vocal fatigue increases vocal fold tension, which in turn causes the mean vocal $f_o$ for each trial to drift upwards in later trials [123–125]. In behavioral data analysis, a control experiment with the same number of trials as an adaptive experiment is carried out and each trial mean adaptive $f_o$ magnitude is subtracted from the adaptive experiment responses to account for this drift [68,69]. In contrast, LaDIVA does not simulate accumulated vocal fatigue over multiple trial repetitions. For each trial, LaDIVA simulated VF dynamics starting from the same initial conditions and there were no carry over components of VF characteristics related to vocal fatigue. In future versions of LaDIVA, the VF model could incorporate features that represent effects accumulated muscle fatigue (i.e., increased tension in intrinsic VF musculature).

## Future directions

LaDIVA can be used to expand the understanding of the physiology of human phonation and to elucidate the origins of specific disorders affecting voice that have or are speculated to have neurological origins (i.e., laryngeal dystonia, hypokinetic dysarthria, vocal hyperfunction). The improvements introduced in LaDIVA provide practical benefits for the investigation of key aspects of the auditory-motor function involved in human phonation. We foresee the application of LaDIVA to simulate behavior of laryngeal motor control for speakers with typical vocal function as well as speakers with voice disorders. LaDIVA can thus provide informative clinical insights for generating hypotheses for future behavioral studies. Moreover, LaDIVA can be used to fit individual participant data. This subject-specific parameterization can provide better understanding of different conditions and individual variability seen in previous behavioral studies [69,70].

Behavioral data collected from cohorts with voice disorders can be used to validate the hypothesis-driven modeling efforts for voice disorders, in a manner similar to the simulations carried out in our model validation. One such example is modeling unilateral VF paralysis (UVFP). UVFP is classically viewed as an isolated peripheral motor condition [126,127]. However, a recent pilot study carried out in a cohort with UVFP has suggested that this isolated peripheral injury to the larynx may have important consequences that impact central auditory processing [58]. Individuals with UVFP were found to generate significantly smaller compensatory responses to reflexive vocal $f_o$ perturbations, compared to responses from adults with typical speech. However, the study findings are challenged by the sex-mismatch in the group of individuals with UVFP and control participants (i.e., the majority of participants in UVFP group were female and the majority of participants in control group were male). Unfortunately, sex is an external factor affecting vocal $f_o$ control, and male speakers have been observed to produce significantly larger vocal $f_o$ responses in prior work [128]. Thus, there is not sufficient evidence to conclude whether the group difference found was due to a sex-mismatch or due to an underlying impairment in vocal motor control in individuals with UVFP. LaDIVA's incorporation of both neural motor control and biomechanical mechanisms of vocal

production could allow for a deeper understanding of UVFP. The extended BCM could be modified to mimic the physiology of a larynx with UVFP in LaDIVA. A patient-specific UVFP-based LaDIVA model, with an asymmetric VF model with reduced/null motor control signaling to one VF, may provide insights about the laryngeal motor control of the disorder [49,53,129]. The model could provide simulated patterns of CT and TA muscle activations and subglottal pressure in the process of simulating responses to reflexive $f_o$ auditory perturbations, which would be helpful in understanding the underlying mechanisms of UVFP.

The main objective of the current implementation of LaDIVA was to assess the validity of a computational model of neural control of laryngeal function with neural and biomechanical components. We have opted to use a combination with minimal computational complexity, providing near real-time simulations of vocalizations. Thus, LaDIVA uses a low-order lumped-element model to describe the biomechanics of phonation that does not model the complexities of three-dimensional geometry characteristics of the VFs provided by higher-order VF models [81]. In future iterations of LaDIVA we hope to incorporate more accurate models that could increase the accuracy of VF modeling of LaDIVA. Another simplification in LaDIVA was maintaining subglottal pressure at 800 Pa through all simulations. This was an intention choice to characterize the effects of different laryngeal muscle activation configurations on vocal $f_o$ and SPL outputs in a controlled manner. However, future work should be carried out to improve the forward model solution optimization to handle instances in which multiple vocal $f_o$ and SPL output solutions are available for a selected combination of $a_{CT}$, $a_{TA}$, and $P_s$ in a way that does not cause discontinuities in vocal $f_o$ values across time steps in model simulations. Currently, there is a future implementation underway considering additional inputs and outputs to extend the scope of the model to incorporate all five intrinsic muscle activation variations (i.e., CT, TA, LCA, PCA, and IA) and vocal quality-based modifications.

## Conclusion

In this study, we have introduced and validated LaDIVA, a new neurocomputational model with physiologically based laryngeal motor control. LaDIVA combines the DIVA model for neural motor control with a lumped-element physics-based vocal fold model. LaDIVA simulations were qualitatively and quantitatively compared and contrasted against adaptive and reflexive behavioral responses to vocal $f_o$ auditory feedback perturbations collected in individuals with typical speech. In addition, simulations of different natural intonation patterns for a running speech utterance were successfully performed. LaDIVA allows for investigation of comprehensive parameter variations, complex stationary and dynamic perturbations that are difficult to assess psychophysically, and causal effects, thus providing a tool for advancing the understanding of typical and disordered voice production.

## Materials and methods

### Notation

**Mobility space** $(x, \dot{x})$ represents positions and changes in positions of a physical structure being controlled. In LaDIVA, mobility space is defined by 14 variables (see *Fig 1*; mobility space pathways denoted in orange). Ten variables are used to represent the vocal tract, each variable defining a specific vocal tract articulatory composition ($x_1$–$x_{10}$). Three variables represent the larynx, each variable defining activation levels of three intrinsic laryngeal muscles ($a_{CT}$, $a_{TA}$, $a_{LC}$). One variable represents the respiratory system, defining the subglottal pressure applied on the larynx via the lungs ($P_s$). The first ten variables representing the vocal tract are preserved from DIVA model, and the variables representing the larynx and respiratory system are modified in LaDIVA. More details are provided in the *Extended Body-Cover Model* section.

Mobility space variables are represented by 1) the symbol $\mathbf{x}$ if referring to a mobility space state, and 2) the symbol $\dot{\mathbf{x}}$ if referring to the change in the mobility space state.

**Task space** ($\mathbf{y}$, $\dot{\mathbf{y}}$) represents sensory inputs in auditory and somatosensory domains. In LaDIVA, auditory task space is represented by five variables (see *Fig 1*; auditory task space pathways denoted in purple). Three variables represent the vocal tract and are defined as the first three resonant frequencies of the vocal tract. These are referred to as first, second, and third vowel formants ($F_1$, $F_2$, and $F_3$). Two variables represent the larynx and are defined as the vocal fundamental frequency ($f_o$; the acoustic correlate of vocal pitch), and the vocal sound pressure level (SPL; the acoustic correlate of loudness). In LaDIVA, somatosensory task space is represented by seven variables, which are preserved from DIVA (see *Fig 1*; somatosensory task space pathways denoted in green). Six variables represent the vocal tract and are defined as constriction levels of six different places of constriction along the vocal tract ($S_1 - S_6$). One variable represents the larynx and is defined as the constriction at the glottis (referred to as the glottal constriction in DIVA) in somatosensory task space. Task space variables are generally represented by 1) the symbol $\mathbf{y}$ if referring to a task space state, and 2) the symbol $\dot{\mathbf{y}}$ if referring to the change in task space state. More specifically, auditory and somatosensory task space variables are referred to as $y_{aud}$ and $y_{somat}$, respectively.

**Targets $r_\alpha(\beta, t)$**, where $\alpha$ = {motor, aud, somat} and $\beta$ = {x, y}, represent the specific references for each sound unit, provided in the speech sound map in the brain (see Fig 1; outputs of speech sound map). Targets are defined as auditory and somatosensory task space reference targets for a given speech sound. The initial states of mobility state variables are set to zero for untuned targets. Through several iterations of simulations, the mobility space variables can be *tuned* to produce the desired auditory and somatosensory task space outcomes. The tuning process involves identifying the auditory and somatosensory feedback errors from desired auditory and somatosensory targets and produced auditory and somatosensory outputs of the system in current iteration and incorporating corrective motor programs to subsequent productions from the system using inverse mapping capabilities. See Inverse Mapping subsection for more details. These tuned targets can be saved as mobility space variables. As reference targets vary over time, they are defined as time series of minimum and maximum target values that each domain variable can operate within (e.g., $r_{audmin}(y,t)$ and $r_{audmax}(y,t)$). From these maximum and minimum values, the value closest to the domain variable at each time step is considered the target for the purpose of error calculation.

In LaDIVA, similar to DIVA, targets are specified in auditory and somatosensory task space variables when initially loaded for simulations. For all simulations using the sustained vowel /a/, the vowel /a/ was tuned to obtain the desired auditory and somatosensory reference targets by tuning the mobility space variables. For the sustained vowel /a/, the vocal $f_o$ target maximum and minimum are defined to be + 5 and—5 cents from baseline $f_o$ value of production. For the sentence 'Buy Bobby a puppy', first the sentence was tuned to obtain the desired auditory and somatosensory reference targets in articulatory domain by setting the vocal $f_o$ target to be monotonic (i.e., $f_o$ = 134 Hz) across the utterance. Then, for each of the four prosodic contours, the sentence was separately tuned to obtain the desired auditory reference target defining each prosodic contour. For prosodic contours, both vocal $f_o$ target maximum and minimum were set to the baseline $f_o$ contour of each intonation pattern, thus a point trajectory was considered instead of a region.

In the current version of LaDIVA, the perceptual acuity (i.e., measured by just noticeable difference (JND) between two values in a task space variable) is considered to be identical to the reference target of that task space variable. For example, for the sustained vowel /a/, the vocal $f_o$ target maximum and minimum are defined to be + 5 and—5 cents from the baseline $f_o$ value of the production. Thus, an error larger than 5 cents from baseline vocal $f_o$ will be

detected and corrected by the control system. The just noticeable difference is also set to be 5 cents. In future versions of LaDIVA, acuity in each task space can be differed to provide additional variability to the targets.

**Intrinsic delays $\tau_{variable}$,** represent the intrinsic delay parameters defined in DIVA to mimic neural and biomechanical processing delays. The propagation delay of signal transmission in cortico-cortical connections between premotor cortex and motor cortex as well as between auditory/somatosensory areas and motor cortex is set to 5 ms. The time it takes for a motor command to have effects on articulatory mechanisms is set to 50 ms. The time it takes an acoustic signal transduced by cochlear to make its way to the auditory cortical areas (i.e., superior temporal cortex; auditory state map in DIVA) is set to 50 ms. The time it takes for somatosensory feedback from peripheral regions to reach higher order somatosensory cortical areas (i.e., inferior parietal cortex; somatosensory state map in DIVA) is set to 20 ms. The rest of the delays in DIVA correspond to 'learned' or 'necessary' delays for accurate representation of neural signal calculations. The learned delays between premotor cortex (speech sound map in DIVA) and auditory and somatosensory areas (auditory and somatosensory error maps) are set to 55 ms and 25 ms, respectively, to make the auditory and somatosensory expectation signals arrive at the error maps at the same time that the corresponding auditory and somatosensory state signals do, such that the error signals are computed correctly. The learned delays between premotor cortex and motor cortex (for the pathway through the cerebellum) are similarly set to 55 ms and 25 ms, respectively for auditory and somatosensory components, to make the learning signals arrive at the motor cortex at the same time that the corresponding feedback corrective signal such that the correct portion of the feedforward command is adapted. These 'learned' delays are necessary for the correct behavior of the model but they could be implemented in many different ways (e.g., differentially delaying the auditory and somatosensory areas' projections to motor cortex to make the somatosensory and auditory error signals arrive at the same time despite different intrinsic delays, so that the rest of the 'learned' delays to the motor cortex need not differentiate between the auditory and somatosensory signals). The equations in the following sections include delay parameters discussed above in more detail. Note that the delays above are those specified in the DIVA Simulink version (2017), which were used for implementation of LaDIVA. Please refer to Guenther, Ghosh [21] for a detailed description of DIVA intrinsic delays and the neurobiological literature supporting the selections.

## Control Modules in LaDIVA

**Auditory feedback controller.**   In LaDIVA, the auditory feedback controller calculates the difference between the *reference auditory target*, $r_{aud}(y,t)$, and the *current auditory output*, $C_{aud}(y)$, to calculate the *auditory error signal*, $e_{aud}(y)$ (Eq 1). The Jacobian inverse mapping, $J(x)^{-1}$, translates the auditory error from auditory task space to mobility space (i.e., $e_{aud}(y) \rightarrow e_{aud}(x)$; Eq 2). A fraction of the translated error is then carried forward as the output of the auditory feedback controller, $FB_{aud}(\dot{x})$, to be added as a corrective command to the feedforward controller. To achieve this fractional addition, the translated error is multiplied by a gain factor (*auditory feedback gain*; $0 < g_{fb\_aud} < 1$). See Eq 3.

$$e_{aud}(y, t) = r_{aud}(y, t) - C_{aud}(y, t) \tag{1}$$

$$e_{aud}(x, t - \tau_{AuM}) = J(x)^{-1} * e_{aud}(y, t - \tau_{AuM}) \tag{2}$$

$$FB_{aud}(\dot{x}, t) = g_{fb\_aud} * e_{aud}(x, t - \tau_{AuM}) \tag{3}$$

Here, reference auditory target = $r_{aud}(y,t)$, current auditory output = $C_{aud}(y,t)$, auditory error signal in auditory task space = $e_{aud}(y,t-\tau_{AuM})$, auditory error signal in mobility space = $e_{aud}(x,t-\tau_{AuM})$, Jacobian inverse mapping = $J(x)^{-1}$, auditory feedback gain = $g_{fb\_aud}$ and output of the auditory feedback controller = $FB_{aud}(\dot{x},t)$. $\tau_{AuM}$ (set to 5 ms) is the long range cortico-cortical signal transmission delay between superior temporal cortex (i.e., auditory error map) and motor cortex (i.e., articulatory velocity position map). Note that $FB_{aud}(\dot{x},t)$ is a 14-n array containing the rate of change in mobility space variables. Thus, the exact corrective command, $FB_{aud}(x,t) = \int FB_{aud}(\dot{x},t)\,dt$.

**Somatosensory feedback controller.** Similarly, the somatosensory feedback controller in LaDIVA calculates the difference between the *reference somatosensory target*, $r_{somat}(y,t)$, and the *current somatosensory output*, $C_{somat}(y)$, to calculate the *somatosensory error signal*, $e_{somat}(y)$ (Eq 4). The Jacobian inverse mapping, $J(x)^{-1}$, translates the somatosensory error from somatosensory task space to mobility space (i.e., $e_{somat}(y) \rightarrow e_{somat}(x)$; Eq 5). A fraction of the translated error is then carried forward as the output of the somatosensory feedback controller to be added as a corrective command to the feedforward controller. To achieve this fractional addition, the translated error is multiplied by a gain factor (*somatosensory feedback gain*; $0 < g_{fb\_somat} < 1$). See Eq 6.

$$e_{somat}(y,t) = r_{somat}(y,t) - C_{somat}(y,t) \tag{4}$$

$$e_{somat}(x,t - \tau_{SoM}) = J(x)^{-1} * e_{somat}(y,t - \tau_{SoM}) \tag{5}$$

$$FB_{somat}(\dot{x},t) = g_{fb\_somat} * e_{somat}(x,t - \tau_{SoM}) \tag{6}$$

Here, reference somatosensory target = $r_{somat}(y,t)$, current somatosensory output = $C_{somat}(y,t)$, somatosensory error signal in somatosensory task space = $e_{somat}(y,t-\tau_{SoM})$, somatosensory error signal in mobility space = $e_{somat}(x,t-\tau_{SoM})$, Jacobian inverse mapping = $J(x)^{-1}$, somatosensory feedback gain = $g_{fb\_somat}$, and output of the somatosensory feedback controller = $FB_{somat}(\dot{x})$. $\tau_{SoM}$ (set to 5 ms) is the long range cortico-cortical signal transmission delay between inferior parietal cortex (i.e., somatosensory error map) and motor cortex (i.e., articulatory velocity position map). Note that the exact corrective command, $FB_{somat}(x,t) = \int FB_{somat}(\dot{x},t)dt$.

**Feedforward controller.** In LaDIVA, the feedforward controller calculates the *feedforward motor command* $FF_{motor}(x,t)$ by multiplying the *reference mobility target*, $r_{motor}(x,t)$ by a weighting parameter $w_{motor}(x)$ and the total feedforward gain $g_{ff_{total}}$. See Eq 8. The weighting parameter $w_{motor}(x)$ is calculated by considering a fraction of the corrective motor commands from auditory and somatosensory feedback controllers (i.e., $e_{aud}(x,t)+e_{somat}(x,t)$). See Eq 7A. The feedforward learning rate is defined to quantify the fraction of feedback corrective commands incorporated into *subsequent motor output* $C_{motor}^{i+1}(x,t)$, (*Feedforward learning rate*; $0 < \lambda_{ff} < 1$). See Eq 7B. Note that the original DIVA model has an additional gain parameter defined to represent total feedforward gain (*total feedforward gain*; $0 < g_{ff_{total}} < 1$) that should not be confused with $\lambda_{ff}$. Given that an adult system should possess fully tuned motor programs, the feedforward gain was set to its maximum value ($g_{ff} = 1$). Unless there is an anatomical change or sensory deficit that affects speech production, the feedforward gain is not expected to differ across time courses relevant to behavioral paradigms [8]. The current motor output $C_{motor}^i(x,t)$ is a combination of the feedforward motor command $FF_{motor}(x,t)$ and a fraction of the corrective motor commands from auditory and somatosensory feedback controllers. See Eq 9. The total feedback gain is defined to quantify the fraction of the combined

auditory and somatosensory feedback corrective commands incorporated into *subsequent motor output* $C_{motor}^{i+1}(x, t)$ (*total feedback gain*; $0 < g_{fb_{total}} < 1$). For LaDIVA simulations, $g_{fb_{total}}$ was set to 1.

$$e_{motor}(x, t) = e_{aud}(x, t) + e_{somat}(x, t) \tag{7A}$$

$$\Delta w_{motor}(x) = \lambda_{ff} * e_{motor}(x, t) * Cb(x, t - \tau_{learnt}) \tag{7B}$$

$$w_{motor}(x) = w_{motor}(x) + \Delta w_{motor}(x) \tag{7C}$$

$$FF_{motor}(x, t) = g_{ff_{total}} * r_{motor}(t - \tau_{PreM}) * w_{motor}(x) \tag{8}$$

$$C_{motor}^{i+1}(x, t) = FF_{motor}(x, t) + g_{fb_{total}} * \int (FB_{aud}(\dot{x}, t) + FB_{somat}(\dot{x}, t))\, dt \tag{9}$$

Here reference mobility target = $r_{motor}(x,t)$, subsequent motor output = $C_{motor}^{i+1}(x, t)$, feedforward motor command error = $e_{motor}(x,t)$, output of the feedforward motor controller = $FF_{motor}(x,t)$, output of the auditory feedback controller = $FB_{aud}(\dot{x}, t)$, output of the somatosensory feedback controller = $FB_{somat}(\dot{x}, t)$. For LaDIVA, auditory task space variables: $f_o$ and SPL, mobility space variables: $a_{CT}$, $a_{TA}$, $a_{LC}$, and $P_s$, and the somatosensory task space variable: glottal constriction, are the relevant parameters. The learned delays $\tau_{learnt}$ between premotor cortex and motor cortex (for the pathway through the cerebellum) are set to 55 ms and 25 ms, respectively, for auditory and somatosensory components to make the learning signals arrive at the motor cortex at the same time that the corresponding feedback corrective signal such that the correct portion of the feedforward command is adapted. $\tau_{PreM}$ (set to 5 ms) is the long range cortico-cortical signal transmission delay between premotor cortex (i.e., speech sound map) and motor cortex (i.e., articulatory velocity position map).

**Forward mapping *F{x}*.**    The forward model in LaDIVA, denoted by F{~}, predicts the auditory consequences, $C_{aud}(y,t)$, of the CT and TA laryngeal muscle activations and subglottal pressure buildup due to respiratory function. See Eq 10. Although somatosensory consequences of motor commands, $C_{somat}(y,t)$, can also be predicted, the current version of LaDIVA focuses on auditory consequences. The forward mapping is a consolidation of 1) conversion of current motor output $C_{motor}(x,t-\tau_{MoAc})$, to positions of the articulators Artic(x,t), 2) the conversion of positions of the articulators to the resultant to acoustic output Acoustic(t) as derived by the acoustic synthesizer in DIVA, and 3) the conversion of the acoustic signal to the current auditory output at the cochlear $C_{aud}(y,t)$. Thus, Eq 10 can be further broken down to three equations. See Eqs 10A–10C.

$$C_{aud}(y, t) = F\{C_{motor}^i(x, t - \tau_{MoAc} - \tau_{AcAu})\} \tag{10}$$

$$Artic(x, t) = F_{MoAr}\{C_{motor}^i(x, t - \tau_{MoAc})\} \tag{10A}$$

$$Acoustic(t) = Synthesizer\{Artic(x, t)\} + Pert(t) \tag{10B}$$

$$C_{aud}(y, t) = F_{AcAu}\{Acoustic(t - \tau_{AcAu})\} \tag{10C}$$

If any auditory perturbation is added to the system, it will be added to the acoustic signal as per Eq 10B. Here, $\tau_{MoAc}$ is the time it takes for the motor command to have its effects on the

articulatory mechanisms. $\tau_{AcAu}$ is the time it takes an acoustic signal transduced by the cochlea to make its way to the auditory cortical areas.

**Jacobian Inverse Mapping $J(x)^{-1}$.** The Jacobian inverse mapping in LaDIVA, denoted by $J(x)^{-1}$, converts the auditory error signals from auditory task space to mobility space (i.e., $C_{aud}(y,t)$ to $C_{aud}(x,t)$). Similarly, somatosensory error signals in somatosensory task space are converted to mobility task space (i.e., $C_{somat}(y,t)$ to $C_{somat}(x,t)$). In the current version of LaDIVA, we focus on auditory error signals. Eqs 11 – 14 summarize the computations involved in generating the Jacobian inverse mapping required to convert the auditory error signals from auditory task space to mobility space corrective motor commands.

$$C^i_{\text{motor}}(x + \Delta x, t - \tau_{learned}) = C^i_{\text{motor}}(x, t - \tau_{learned}) + \xi * I(x) \tag{11}$$

$$e_{aud}\left(\frac{y}{\Delta x}\right) = F\left\{C^i_{\text{motor}}(x + \Delta x, t - \tau_{learned})\right\} - C_{aud}(y, t) \tag{12}$$

$$J(y) = e_{aud}\left(\frac{y}{\Delta x}\right) * e_{aud}\left(\frac{y}{\Delta x}\right)^T \tag{13}$$

$$J(x)^{-1} = \xi * I(x) * e_{aud}(\dot{y}) * [J(y) + \gamma * \xi^2 * I(y)]^+ \tag{14}$$

$$e_{aud}(x, t - \tau_{AuM}) = J(x)^{-1} * e_{aud}(y, t - \tau_{AuM}) \tag{2}$$

Here, $F\{\sim\}$ = forward model in LaDIVA, $C_{aud}(y,t)$ = current auditory consequences, $I(x)$ = identity matrix, $\gamma$ = Jacobian regularization factor, $\xi$ = Jacobian step size, $C_{motor}(x,t)$ = current motor command, $\Delta x$ = Jacobian step size change in mobility space, $e_{aud}\left(\frac{y}{\Delta x}\right)$ = auditory task space change for a Jacobian step size change in mobility space, $J(y)$ = Jacobian matrix, $J(x)^{-1}$ = Jacobian inverse mapping, $[\sim]^+$ = Moore Penrose pseudo inverse, $e_{aud}(y,t)$ = auditory error signal in auditory task space, $e_{aud}(x,t)$ = auditory error signal in mobility space. $\tau_{learned}$ is the delay within the motor cortex between the current motor position signal and the Jacobian inverse-map. $\tau_{learned}$ is set to 55 ms make the current articulator position signal arrive at the inverse-map at the same time that the corresponding error signals such that the inverse projection is computed based on the motor configuration of the articulators at the time of the error generating production.

## Modification to extended BCM

**Overview of extended BCM.** To incorporate motor control of glottal function into DIVA, biomechanical low-dimensional lumped-element modeling of vocal fold (VF) oscillations was applied. For this, an extended version of the well-established body-cover model (BCM) developed by Story and Titze [46] resembling internal layered tissue structure in the VFs was chosen. The extended version of the BCM introduced by Zañartu, Galindo [49], which contains the original BCM mathematical equations as well as added features including a revised glottal flow model with a posterior glottal gap, was used. The BCM formulation allows for accurate description and simulation of many fundamental aspects of vocal function, at a low computational cost [45]. These include flow-induced self-sustained oscillations, the vertical mucosal surface wave, and fold collision during the closed phase [43,130]. It further considers the three-way interaction at the glottis between sound, flow, and VF tissue; and is compatible with acoustic wave propagation schemes for modeling vocal tract articulation [131]. In the extended BCM model, the effects of a posterior gap on the VF tissue dynamics are also taken into account, which is crucial for modeling vocal hyperfunction [49].

For the implementation in this paper, the DIVA acoustic synthesizer was applied to generate the acoustic signal outputs of the model (see *Materials and Methods*: *acoustic synthesizer* section). However, it should be noted that DIVA does not contain biomechanically-relevant source-filter interactions in its acoustic synthesizer. Intrinsic laryngeal muscles control the VF configurations in the BCM. The thyroarytenoid (TA) muscle comprises the body of the VFs and contributes to changing the tension and length of the VFs. The cricothyroid (CT) muscle tilts the thyroid cartilage forward, which in turn tightens and stretches the VFs. The interarytenoid (IA) and lateral cricoarytenoid (LCA) muscles work in conjunction to adduct the glottis. The opposite action is performed by the posterior cricoarytenoid (PCA) muscle solely. Intrinsic laryngeal muscles also control VF posturing and tissue properties. In the BCM, VF adjustment (i.e., controlled modification of geometrical and biomechanical model parameters) is obtained using a simplified muscle activation scheme based upon a set of physiological rules [46,48]. The defined set of rules allow for capturing the essential effects on vocal function during phonation due to the activation of intrinsic musculature. The rules consider normalized activation levels of CT ($a_{CT}$) and TA ($a_{TA}$) muscles, whereas the resulting adductor effect due to both the LCA and PCA is represented through a single activation level ($a_{LC}$). Subglottal pressure ($P_s$) is not controlled by the rules and it is a separate input parameter that controls the driving force of the self-sustained model and affects the resulting amplitude and pitch. In the current implementation of LaDIVA we have mainly focused on the activations of CT and TA intrinsic muscles for the production of acoustic outcomes. LCA, PCA, and IA muscle activation is kept constant at an adducted state to simulate voice production via the extended BCM for LaDIVA. Unvoiced productions are handled via the noise generator in DIVA acoustic synthesizer. In future iterations we hope to incorporate all five intrinsic laryngeal muscles (i.e., CT, TA, LCA, PCA, IA) and their activations in LaDIVA.

**Combining DIVA and extended BCM.** The extended BCM requires CT and TA muscle activations ($a_{CT}$, $a_{TA}$) and subglottal pressure ($P_s$) as inputs. These were linked from DIVA vocal fold mobility trajectories. For all simulations, the initial mass, position, velocity, and acceleration of the VFs were specified as per the original BCM formulation [46]. The main output of the extended BCM is the glottal area waveform. However, DIVA requires auditory task space variables as sensory inputs. To generate the auditory targets compatible with DIVA, vocal $f_o$ and radiated SPL signals were generated from the glottal area waveform provided by the extended BCM. Vocal $f_o$ was generated from the glottal area waveform using an autocorrelation algorithm. Radiated SPL at glottis was calculated as $20\log_{10}[rms(Z_v^*(dU_g/dt)/(2x10^{-5})]$, where impedance constant $Z_v$ was set to 45000 kg.m$^{-4}$ and $U_g$ was the glottal flow signal [132]. These derived outputs were used as sensory inputs to DIVA.

**Generating forward maps.** The control loop in the DIVA implementation operates iteratively over 5-ms non-overlapping windows. At each 5-ms time step of an utterance, the motor representations are converted to auditory and somatosensory representations via the forward maps and error signals are calculated and inverse mapped to motor presentations. Thus, connecting DIVA and extended BCM together with a set of auditory task space parameters and motor mobility space parameters at their connection interface meant that simulations were required to be run every 5 ms across the two systems. Firstly, this limited the near real-time simulations of LaDIVA. Secondly, a 5-ms simulation in extended BCM was not sufficient to generate a glottal airflow signal stable enough to calculate a vocal $f_o$ (i.e., due to vocal variability at the start of production in extended BCM). See *Acoustic Synthesizer* section below $f_o$ more details. Thus, we generated a forward mapping for all combinations of motor mobility space variables (i.e., $a_{CT}$, $a_{TA}$, and $P_s$) to auditory task space variables (vocal $f_o$ and SPL), using model simulations of the extended BCM for all possible combinations of input parameters; $a_{CT}$ (range: 0–1, initial step size: 0.02, smoothed step size: 0.001), $a_{TA}$ (range: 0–1, initial step size:

0.02, smoothed step size: 0.001), and $P_s$ (range 10–2010 Pa, initial step size: 100 Pa, smoothed step size: 5 Pa). Linear interpolation was used to smooth the forward map outputs. This forward mapping was connected to the DIVA model to replicate vocal fold biomechanical parameters in the feedforward controller (see *Fig 1*). In a similar manner, the Jacobian inverse functions were modified to refer to the new forward mapping added for vocal fold parameters in order to conduct the inverse mapping required in the error correction processes in the auditory and somatosensory feedback controllers [22] (See *Fig 1*).

**Behavioral study dataset.** In order to validate LaDIVA, an experimental dataset previously collected from 20 female adults with typical speech and no neurological, speech, or hearing disorders was used [68]. The participants underwent reflexive and adaptive vocal $f_o$ perturbation paradigms, and the group responses for each paradigm were considered as the reference standard in the current study to fit the simulated responses generated via the new model implementation. For both vocal $f_o$ reflexive and adaption paradigms, participants were asked to repeatedly produce the sustained vowel /α/ for 108 trials. For the vocal $f_o$ reflexive paradigm, auditory feedback for the vocal production was delivered unaltered for 75% of the trials. For a randomly selected 25% of the trials, auditory feedback presented via headphones to the participants was altered in real-time. The vocal $f_o$ was increased by +100 cents using pitch shifting hardware (Eclipse V4 Harmonizer; Eclipse, Little Ferry, NJ), and presented with a jittered perturbation onset of 500–1000 ms from vocal onset (see *Fig 3A*). For the vocal $f_o$ adaptive paradigm, the first 24 trials of the paradigm (termed *baseline phase*) had unperturbed auditory feedback provided to the participant. The next 30 trials of the paradigm (termed *ramp phase*) had each trial's auditory feedback shifted by an increasing step size of 3.33 cents from 0 to +100 cents. For the next 30 trials (termed *hold phase*), the auditory feedback perturbation was held steady at +100 cents in each trial. The last 24 trials (termed *after effect phase*) contained unperturbed auditory feedback (see *Fig 3B*). The vocal $f_o$ of the auditory feedback signal was shifted prior to vocal onset and the perturbation was sustained until the end of the trial.

**Model simulations.** LaDIVA was implemented in MATLAB based on the DIVA implementation using the Simulink platform. All simulations for LaDIVA were carried out via MATLAB scripts and Simulink (Mathworks, Natick, MA, Version R2018a). Reflexive and adaptive vocal $f_o$ perturbation paradigms identical to those in the behavioral dataset were simulated for LaDIVA with initial muscle activations and subglottal pressure set to replicate a male voice with vocal $f_o$ = 134 Hz (achieved via four initial cases of laryngeal muscle activation levels; see *Table 4*).

**Acoustic synthesizer.** The incorporation of the extended BCM in LaDIVA follows the same principle in order to be compatible with the DIVA control loop and acoustic synthesizer. See *Fig 1* and *Table 1* for the LaDIVA control loop architecture and controlled parameter list, respectively. The control loop in the DIVA implementation is decoupled from its acoustic synthesizer such that the control loop operates iteratively over 5 ms non-overlapping windows. At each 5-ms time step, the motor representations are converted to auditory and somatosensory representations via the forward maps and error signals are calculated and inverse mapped to motor presentations. See Eq 10 in the *Forward mapping* section above. At the end of the full simulation (e.g., an utterance of total duration of 3050 ms for the sustained vowel /a/), the motor representation time series is applied to the acoustic synthesizer to generate the time varying acoustic signal. See Eqs 10A and 10B.

The reasoning for not calculating the acoustic signal per each 5 ms time window is as follows. According to the source-filter theory, the glottal air volume velocity (glottal air flow) is the source of phonation [133]. The glottal airflow signal is filtered through the vocal tract model for modulating the spectral information (e.g., enhancing spectral formant regions or

deemphasizing anti-formant regions; [47]). In the latest version of the DIVA model (dated Oct 22, 2017), a static signal for the glottal air flow is simulated via the glottal function in Liljencrantz-Fant glottal flow model [77], which is filtered via the time-varying vocal tract filter of DIVA speech synthesizer. For the LaDIVA implementation, we inputted trajectories of CT and TA muscle activation and subglottal pressure signals as control signals to the extended BCM to simulate VF dynamics for a full speech utterance (i.e., 3050 ms duration) that in turn generated a time-varying glottal area waveform. The derivative of this waveform is calculated to generate the glottal airflow signal, that is then filtered by the time-varying DIVA vocal tract filter to generate the resulting acoustic output of LaDIVA. The VF dynamics require initial kinematics of VFs (i.e., position, velocity, and acceleration) to be defined at the beginning of simulation and outputs end kinematics of VFs at the end of each simulation. If the muscle activations and subglottal pressure were fed to extended BCM every 5 ms, the simulation would reset the VF kinematics to initial state at the beginning of each 5 ms, which would generate discontinuities in VF movement and thereby discontinuities in the derived glottal airflow signal. Moreover, a 5 ms window is not sufficient for the extended BCM to generate a stable glottal airflow signal to calculate vocal $f_o$. Thus, for the purposes of acoustic synthesis, the glottal area waveform is not derived for 5 ms time steps of the vocal utterance and parsed together. Instead, the glottal area and air flow signals for a complete vocal utterance are generated via the extended BCM by providing a time series of muscle activations and subglottal pressure and simulating VF dynamics and thereby a glottal airflow waveform for the total utterance.

## Supporting information

**S1 Audio. Audio recordings of original and synthesized utterances of different prosodic contours for the sentence 'Buy Bobby a Puppy'.**
(RAR)

**S1 Text. Detailed calculation of compensatory response magnitude for the vocal $f_o$ reflexive perturbation of +100 cents when full auditory feedback compensation is considered.**
S1 Text: Simulation responses of the LaDIVA model for the vocal fo reflexive paradigm plotted over a longer period of time.
(DOCX)

**S2 Text. Vocal $f_o$ adaptive response when production noise is included in intrinsic laryngeal muscle activation levels.**
(DOCX)

## Acknowledgments

We acknowledge Alfonso Nieto-Castañón and Jason Tourville for providing support related to original DIVA model source code. We acknowledge *Rodrigo* Manríquez for contributing to the software in the generation of vocal $f_o$ outputs from glottal airflow signals.

## Author Contributions

**Conceptualization:** Cara E. Stepp, Matías Zañartu.

**Funding acquisition:** Cara E. Stepp, Matías Zañartu.

**Investigation:** Hasini R. Weerathunge, Cara E. Stepp.

**Methodology:** Hasini R. Weerathunge, Matías Zañartu.

**Software:** Hasini R. Weerathunge, Gabriel A. Alzamendi, Gabriel J. Cler, Frank H. Guenther.

**Supervision:** Cara E. Stepp, Matías Zañartu.

**Visualization:** Hasini R. Weerathunge.

**Writing – original draft:** Hasini R. Weerathunge, Gabriel A. Alzamendi, Gabriel J. Cler, Matías Zañartu.

**Writing – review & editing:** Hasini R. Weerathunge, Frank H. Guenther, Cara E. Stepp, Matías Zañartu.

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
