## [Decision Letter · Decision Letter 0]

1 Dec 2021

Dear Ms. Weerathunge,

Thank you very much for submitting your manuscript "LaDIVA: A neurocomputational model providing laryngeal motor control for speech acquisition and production" for consideration at PLOS Computational Biology.

As with all papers reviewed by the journal, your manuscript was reviewed by members of the editorial board and by several independent reviewers. In light of the reviews (below this email), we would like to invite the resubmission of a significantly-revised version that takes into account the reviewers' comments.

Dear Authors,

Your paper has now been reviewed by two experts in the field. They have both made extensive comments on your manuscript. Clearly the technical issues that were raised need to be addressed. Reviewer 2 has also raised important concerns on the scope of your model, more specifically that it does not address the principal deficits that are observed in voice disorders (phonation) and that it lacks a neurobiological component. Both of these were promised in the introduction. Optimally you could address these shortcomings by actually expanding your modeling efforts but, alternatively, by lowering expectations. You might be able to find the correct middle ground in your revised manuscript.

I will look at your reply carefully.

Best wishes,

Frederic Theunissen

We cannot make any decision about publication until we have seen the revised manuscript and your response to the reviewers' comments. Your revised manuscript is also likely to be sent to reviewers for further evaluation.

Sincerely,

Frédéric E. Theunissen

Associate Editor

PLOS Computational Biology

Thomas Serre

Deputy Editor

PLOS Computational Biology

Dear Authors,

Your paper has now been reviewed by two experts in the field. They have both made extensive comments on your manuscript. Clearly the technical issues that were raised need to be addressed. Reviewer 2 has also raised important concerns on the scope of your model, more specifically that it does not address the principal deficits that are observed in voice disorders (phonation) and that it lacks a neurobiological component. Both of these were promised in the introduction. Optimally you could address these shortcomings by actually expanding your modeling efforts but, alternatively, by lowering expectations. You might be able to find the correct middle ground in your revised manuscript.

I will look at your reply carefully.

Best wishes,

Frederic Theunissen

Reviewer's Responses to Questions

**Comments to the Authors:**

Reviewer #1: Weerathunge et al combine their existing DIVA model with the extended body-cover model of human vocal fold oscillation. The authors validate their model by simulating two auditory feedback paradigms (reflexive and adaptive), and a time-resolved synthesizer producing prosodic f0 contours.

This is an interesting, well written, important paper with many applications that I wholeheartedly recommend for publication.

I have a few comments:

1. The comparisons between model and behavioral data remain qualitative. Because the authors claim this to be a validation study I recommend to present a more qualitative comparisons including statistics.

2. The experimental data to validate the BCM is mostly based on the Chhetri et al papers using graded muscle stimulation on the canine larynx. I suggest that the authors incorporate a discussion of these papers and also would like to point to recent papers of fluid-structure interaction models of the dog larynx from the Zheng lab (U of Maine, e.g. Geng et al JASA 2021 150,1176-1187). Their more accurate models could be included later in this laDiva as lumped parameter system

3. The predicted muscle activity needs to be experimentally validated in follow-up studies. Modelling alone will never give an answer without validation. The authors should rephrase/town this down in their discussion.

4. The discussion is well balanced and includes current literature, but is very long and the paper would benefit from a more focused condensed discussion.

Specific comments

L108 these citations refer to EMG only

L130; can features be specified?

L166 physical quantity underlying the perceptual

L240 typo: missing citation 60

Fig 3a x-axis should be adjusted to same length as in C.

L273/4 “and the best fit for feedback gain was selected.” And also L292: The comparison between model and behavioral data is very qualitative. A simple difference between area under curve or curve-fit comparison woudl be stronger.

L293 I don’t see why lambda =0.1 has a better fit. A quantitative comparison would be stronger.

L301-302 The similar SPL along isolines of fo is surprising and may be a consequence of the simplicity of the vocal fold model driven at the same Ps.

L307 the data only shows fo and not SPL. Could the authors also include the SPL results in Fig 4B?

L308 see also more recent simulations of these data from the Zheng lab (Geng et al).

L315 This section would benefit from some additional explanation. I fail to see the major point. The model can clearly generate different pitch contours, but this is only a time-resolved implementation of what has been shown before. It is not clear how the prosodic contours were derived from the acoustic input signal. The authors in the section above showed that the TA/CT space is redundant for fo and SPL. Thus multiple activation levels can achieved the same acoustic output. At what positions did they start the simulations for Fig 4?

I would say that the really interesting part is whether the resulting BCM related predicted physiological parameters, such as TA/CT activation, match experimental data.

L323 “simulated outputs fully overlapped with the original recording pitch contours.” The authors don’t include the evidence for this statement.

L338-344 This is maybe true once there is substantial evidence that the predicted levels are correct: such evidence is currently lacking. The authors should tone down this section.

, but these EMG activation levels should be acquired to test the performance.

The discussion is very long and could be condensed

Reviewer #2: This manuscript presents a needed attempt to combine control models of speech production with a biologically realistic model of the vocal folds and larynx. Such a model has the potential to give insight into laryngeal control in typical speakers as well as help identify the mechanisms of vocal disfunction in various neurogenic voice disorders. However, I feel that the current manuscript has a problem with the scope of the work that has been done as it relates to these issues. I outline these in more detail, below. Additionally, there are several technical issues where the description of the model is unclear or incomplete. Lastly, there are some issues of interpretation or discussion of the results that I think are important to clarify.

Model scope:

• I feel there the introduction and motivation for the work doesn’t match what is actually done. The introduction motivates the work primarily from the perspective of enabling a better understanding of voice disorders. However, the vast majority of vocal disorders (to my knowledge, and somewhat supported by looking at the references cited by the authors) are not problems with pitch control per se but structural or neurological problems resulting in phonation quality issues. This isn’t to say that this work isn’t useful—far from it—but the limitations of the current model to address the questions that are used to motivate it needs to be addressed.

• On a similar note, the motivation suggests that there is a need for a model that combines neural control of speech with a realistic phonation model. However, the current model makes no reference no neurobiologically grounded control. While this does not a priori limit the utility of the model, it does not speak to the need established by the authors in the introduction. Though the authors state that LaDIVA is a “neurocomputational model”, I don’t know that I agree—as presented, this seems like a pretty straightforward mathematical control model. That is not necessarily a problem in the abstract, but some discussion of the potential neurological basis of the model seems warranted given the authors stated intent.

• I understand that the point of the current paper is to establish the model as a proof of concept, but I keep coming back to the motivations. Even with pitch control, some effort could be made to show the model may provide useful insights into vocal disorders. This could potentially be from some initial modeling of neurologically-based pitch control problems, or amplitude control problems (since that is also a feature of the model). Or perhaps showing that changes in some of the tunable control parameters could result in phonation quality changes, or really any demonstration that links a change in control to a (pathological) change in behavior. As it stands, the authors have shown the control architecture can control the laryngeal plant model, but not that this is a viable model of neurological impairment in vocal disorders. This is the critical need the authors establish, so some demonstration that the model may fill that need seems warranted.

• To some extent, the ability of the model to replicate compensatory and adaptive behavior is not particularly novel. This control architecture, which is what drives the changes in model behavior, has been shown to be capable of replicating these behaviors previously for auditory perturbations of vowel formants in supralarygneal articulatory control. The novel component here is largely the use of a new plant—the body cover model of the vocal folds—and the needed mapping (Jacobean) between acoustic and plant parameters. As long as that relationship is learnable, the model architecture is known to produce these results. As outlined above, I think the demonstration of this established function would be substantially enhanced by examples of applications to voice disorders.

Missing or incomplete technical information:

• The adaptation component of the model is not described, as far as I can tell. In the traditional DIVA framework, adaptation to feedforward motor plans (motor trajectories) is accomplished by adding a time-shifted version of the feedback-driven compensatory motor commands to the existing motor trajectories, with a gain. No similar process is described in the current manuscript. Indeed, no process at all is described for updating the motor trajectories used by the feedforward control system.

• The text often references a “feedforward gain” (gff). However, this parameter does not appear in the control diagrams of any of the equations. I believe that the feedforward gain is mislabelled as the learning rate (λff) in both the diagrams and equations, as this factor is simply a gain on the motor state error. For example, in equation 8 [FFmotor(x) = λff ∗ emotor(x)], where λff is multiplied by the error to generate the (current) feedforward motor command, that is then summed with the feedback commands in equation 9 to move the plant [Cmotor(x, i + 1) = FFmotor(x) + ∫(FBaud(x) + FBsomat(x)) dt] where C equals the “motor output”. So, this is really the feedforward gain and not the learning rate.

o Now, the process through which the feedforward trajectory (r) is updated maybe similar (and indeed actually rely on λff), but such an update is not specified in the methods.

• The simulation results in Figure 3 A show a latency of around 150 ms in the online compensatory response after the perturbation onset. This suggest that the model has built in delays in the auditory feedback processing pathway. However, no such delays are described in the methods.

• It’s not clear to me how the pitch and amplitude were determined at each time step. The description of acoustic simulation starts on line 763. My understanding here is that that the final acoustic output was synthesized at the end of the trial/utterance based on the history of trajectories generated by the controller. I am wondering how the acoustic signal was generated at each time step, however. For the Maeda plant, this is straightforward, as the plant position can be measured instantaneously (similar for the formants, which are estimated from the transfer function). I would assume that the BCM model takes some time to stabilize vocal production, however (this is suggested by the authors (line 288, where they say there is vocal onset variability up to 40ms in duration). So I am not clear on how the pitch and amplitude values were generated at each time step.

• Line 438: “The JND for vocal fo in LaDIVA was set during all simulations to be significantly smaller compared to the maximum perturbation magnitude utilized in the perturbation paradigms”. There is no description of a JND setting in the methods/description of the model, or any type of perceptual thresholding.

• Line 591:“five variables represent the vocal tract and are defined as constriction levels of six different places of constriction along the vocal tract (S1 − S6 ).” Is it five or six?

• Line 601: “targets are saved as mobility space variables and converted to respective auditory and somatosensory task space reference targets when initially loaded for simulations.” This seems to be a pretty key element, but no description is given for how this “conversion” is accomplished. Some more detail is needed here.

• Line 603: “As reference targets vary over time, they are defined as time series of minimum and maximum target values each domain variable can operate within (e.g., raudmin(y, t) and raudmax(y, t)). For the purpose of error calculation in feedback controllers, the midpoint between minimum and maximum values for each time point is considered as the target for each variable.” So, for all intents and purposes, the targets are point trajectories and not regions. I think this should be made explicit, as it is a departure from the standard implementation of DIVA. This is important, as it is my understanding that in the standard DIVA model (at least as typically described in published papers) errors are defined as the distance to the edge of the target region. So, this is a departure from previous work and may cause different behavior.

• Line 684: “The extended version of the BCM, including a revised glottal flow model with a posterior glottal gap introduced by Zañartu, Galindo (41), was also considered.” I don’t know what this means here—that the extended version was used? That is was not used? How did the results vary between the two versions of the model?

• Line 702: “Intrinsic laryngeal muscles also control VF posturing and tissue properties.” I assume that “intrinsic laryngeal muscles” means all the specified muscles (TA, CT, IA, LCA, PCA), but this should be made clear.

• Line 287: “This window was selected to target the feedforward response as the auditory feedback response is assumed to begin approximately 120 ms after perturbation onset”. The onset in the model is fully a function of the (unspecified) delays built in. So, rather than picking a time window based on human behavioral data, the authors should pick the time window of interest based on the latency response of the model. The window chosen seems like it may be fine for this given the simulations shown in Fig 3A, but I would clarify this point in the text (especially since trajectories for trials in the adaptation experiment are not shown—was the latency the same?).

Interpretation of results:

• I am not sure how informative the parameter setting exploration is for the auditory feedback gain in modeling the compensatory response. As I understand it, this is really and exploration of the RELATIVE gain between the feedforward controller and auditory feedback controller, rather than strictly an exploration of the auditory feedback gain, as presented. This is because, as the auditory feedback controller moves the system away from the specified motor trajectory, the feedforward controller will attempt to move back to that trajectory. So the results of a specific FB gain are only relevant for a fixed FF gain, which is fairly arbitrary. I think a more thorough exploration of these parameters is warranted. Or at a minimum, an acknowledgement that these results only hold for the fixed FF gain used.

• Relatedly, what is the importance of a specific gain fitting the data well (i.e., that “We observed that gfb_aud = 0.7 provided an excellent fit for the behavioral mean vocal fo reflexive response”? Is it just that the model can fit the data? The argument for building the model was that it can help shed light on the neurobiologoical control of vocal pitch, so I would assume that such parameter fitting has some importance beyond just showing the model can fit the data.

• There is a pretty big (and somewhat related) theoretical issue raised by the compensation results. Here, the compensatory response is only a fraction of the applied perturbation, even with an auditory feedback gain of 1 (it looks like the plateau of the response will be around 35 cents relative to the 100 cent perturbation). As the authors point out in the discussion (line 483), such partial compensation has often been attributed to competition between the auditory feedback system and the somatosensory feedback system. However, the current model does not include somatosensory feedback control for pitch and, as such, this result suggests that the partial compensation observed arises directly though competition between the feedback and feedforward control systems (perhaps in addition to competition with somatosensory feedback control). This sort of upends the typical story (and contradicts what the authors themselves write in the discussion), so I believe that some discussion is warranted.

• There is a pretty substantial discussion comparing these results to the simpleDIVA model. I am not sure such a large discussion is warranted, as the models are radically different in both their architecture, and more importantly, intent. The current model is a control model of vocal and supralaryngeal production that is capable of learning. simpleDIVA is, conversely, a way to fit a set of equations to time-series data to estimate gain parameters and is NOT a control model.

• Line 489: “This reasoning [the fact that there is no somatosensory feedback controller in the current model] may explain why LaDIVA’s adaptive simulations do not produce fo contours that fit empirical data as well as its reflexive simulations.” This is possible, but I’m not really convinced that this will solve the problem. Primarily because the largest difference between the simluations of adaptation and adaptation trajectories in real data are that the latter are noisy—they don’t monotonically increase. The difference between the real, noisy adaptation trajectories and the smooth adaptation trajectories produced by the model is clear in Fig 3B. Adding somatosensory feedback will not change this difference See, e.g., Villacorta, V. M., Perkell, J. S., & Guenther, F. H. (2007). Sensorimotor adaptation to feedback perturbations of vowel acoustics and its relation to perception. Journal of Acoustical Society of America, 122(4), 2306–2319. https://doi.org/10.1121/1.2773966, which uses the DIVA framework with somatosensory feedback control but still produces smooth adaptation trajectories. Adding this may change the overall shape of the trajectories to better match the data, which has a steeper rise and faster transition to a plateau, though again the simulations in Villacorta et al. show a similar shape to the simulations here, so it would seem that substantial changes in shape are unlikely. A similar difference in shape can be seen in the washout phase.

More minor points:

• Line 240: missing citation

• Line 78: this is a citation for models of voice production, but citations neither citations 15 nor 16 treat voice production per se (they relate to the DIVA or GODIVA models of (supralaryngeal) control and planning).

**Have the authors made all data and (if applicable) computational code underlying the findings in their manuscript fully available?**

Reviewer #1: Yes

Reviewer #2: **No: **The GitHub repository with the model code is currently private. The authors say that it will be made public on publication.

PLOS authors have the option to publish the peer review history of their article (what does this mean?). If published, this will include your full peer review and any attached files.

Reviewer #1: No

Reviewer #2: No
---

## [Decision Letter · Decision Letter 1]

30 Mar 2022

Dear Ms. Weerathunge,

Thank you very much for submitting your manuscript "LaDIVA: A neurocomputational model providing laryngeal motor control for speech acquisition and production" for consideration at PLOS Computational Biology. As with all papers reviewed by the journal, your manuscript was reviewed by members of the editorial board and by several independent reviewers. The reviewers appreciated the attention to an important topic. Based on the reviews, we are likely to accept this manuscript for publication, providing that you modify the manuscript according to the review recommendations.

Dear authors,

1. Could you take another look at the comments of reviewer 2 on your revision? You will see that they are minor and that the reviewer is supportive of your work.

2. Could you also please publicly post the code for your model?

Best,

Frederic Theunissen

Sincerely,

Frédéric E. Theunissen

Associate Editor

PLOS Computational Biology

Thomas Serre

Deputy Editor

PLOS Computational Biology

[LINK]

Dear authors,

1. Could you please take another look at the comments of reviewer 2 on your revision? You will that they are minor and that the reviewer is supportive of your work.

2. Could you also please publicly post the code for your model?

Best,

Frederic Theunissen

Reviewer's Responses to Questions

**Comments to the Authors:**

Reviewer #1: The authors have professionally considered my concerns and I think this paper will be a great addition to the literature.

Reviewer #2: The authors have done an excellent job of clarifying the technical details of the model. I have a few remaining minor points the authors may wish to address, a few of which remain from the first submission.

I still think there’s perhaps a little too much emphasis but on the “neurobiological” aspect of the model. I do understand that the DIVA model uses a neural network implementation, but the examples of “neurobiological” parameters given in the paper are changes in feedback/learning gains—that is, tunable (and abstract!) mathematical parameters. That is, the authors aren’t suggesting that they can model changes in f0 responses in neurologoical disease through “lesions” to different components in the model or changes to the neural architecture, but by tuning the fixed model parameters (for example: “Thus, the suspected neurological vocal motor control deficits in vocal hyperfunction can be modeled using LaDIVA by decreasing the feedforward learning rate [line 440]”. That’s true, but it doesn’t give any insight into the neurobiology of WHY vocal hyperfunction results in this behavior—it’s simply a change in a tunable control parameter, and you could replicate this behavior in . Again, this isn’t a problem with the work itself—there’s a lot that can be learned from abstracted models! But it seems like there’s some overselling of what is actually happening here.

I also still think that a discussion of the partial compensation seen in the model, even with the auditory feedback gain set to 1, is warranted. I don’t disagree with the authors reply at all—I think we are saying the same thing: namely, that since the corrective auditory feedback-driven motor command is (weighted and) summed with the feedforward command, that the model will maximally produce a partial 50% compensation. This is actually a strong prediction of the model: 1) that partial compensation is driven by competition between feedforward and feedback control systems, and not only between auditory and somatosensory feedback mechanisms, as is often stated in the literature and 2) that compensation should NEVER be above 50%, since this is the ceiling with a feedforward gain of 1.

L320: Relative to the analysis window (“In order to extract the acoustic response driven by feedforward motor commands alone, an analysis window from 40 ms to 120 ms after the vocal onset was selected. This analysis window was selected to target the feedforward motor command based response as auditory feedback is assumed to affect current vocal productions with a latency of 120 ms (63)”)—I still think the critical thing here is that you are choosing this window because you KNOW the model has not yet produced a compensatory response because you are controlling the delays, rather than being based on human behavioral data. E.g., if the model WERE producing a reflexive response in this window, it would not be a good window to assess feedforward learning IN THE MODEL, regardless of delays in real (human) data.

L376: I don’t follow what you mean by “We expect that this methodology can be used in the future to generate stimuli for auditory vocal fo perturbations of running speech.” If this is important, I would suggest explaining this in more detail.

**Have the authors made all data and (if applicable) computational code underlying the findings in their manuscript fully available?**

Reviewer #1: Yes

Reviewer #2: **No: **model code is currently in a private repository that the authors say will be made public on manuscript acceptance.

PLOS authors have the option to publish the peer review history of their article (what does this mean?). If published, this will include your full peer review and any attached files.

Reviewer #1: No

Reviewer #2: No

Figure Files:

Data Requirements:

Reproducibility:

References:

---

## [Editor Report · Decision Letter 2]

2 May 2022

Dear Ms. Weerathunge,

We are pleased to inform you that your manuscript 'LaDIVA: A neurocomputational model providing laryngeal motor control for speech acquisition and production' has been provisionally accepted for publication in PLOS Computational Biology.

Best regards,

Frédéric E. Theunissen

Associate Editor

PLOS Computational Biology

Thomas Serre

Deputy Editor

PLOS Computational Biology

Dear Authors,

Thank you for addressing these last sets of comments.

Best,

Frederic.

---

## [Editor Report · Acceptance letter]

2 Jun 2022

PCOMPBIOL-D-21-01919R2 

LaDIVA: A neurocomputational model providing laryngeal motor control for speech acquisition and production

Dear Dr Weerathunge,

I am pleased to inform you that your manuscript has been formally accepted for publication in PLOS Computational Biology. Your manuscript is now with our production department and you will be notified of the publication date in due course.

With kind regards,

Zita Barta
